# TEXT2GRAPHBENCH: A COMPREHENSIVE BENCHMARK FOR EVALUATING TEXT-INSTRUCTED GRAPH GENERATION WITH LARGE LANGUAGE MODELS

## ABSTRACT

The rise of Large Language Models (LLMs) is driving a paradigm shift in graph generation, from traditional statistical modeling to the emerging paradigm of Text-instructed Graph Generation. However, the development of this research field faces a critical bottleneck: a severe lack of benchmarks specifically designed for this new paradigm. This prevents a reliable and in-depth analysis of the capabilities of existing models. To address this issue, we introduce Text2GraphBench, a comprehensive benchmark designed to evaluate and analyze the performance of large models on this task. At the core of Text2GraphBench is a methodology for benchmark curation and evaluation centered on constraints. For dataset curation, we pioneer a "graph-to-constraint, constraint-to-text" generation pipeline, building a large-scale, multi-domain dataset that ensures every textual instruction corresponds to a precisely verifiable constraint. For the evaluation system, we propose a novel, constraint-based three-dimensional evaluation framework that moves beyond traditional similarity comparisons, assessing generated graphs from the perspectives of Validity, Semantic Fidelity, and Novelty in a thorough and quantifiable manner. We conduct extensive evaluations on a range of mainstream LLMs using Text2GraphBench, and our results provide the first systematic revelation of the current capabilities, strengths, and challenges of these models. We hope that Text2GraphBench will provide the community with a valuable tool to quantify model capabilities and inspire future research. Our datasets, code, and analysis results are fully open-sourced.

## 1 INTRODUCTION

Graphs, as a universal data representation, are widely used in various fields such as biology (Tong et al., 2021), engineering (Mark, 2003), and social sciences (Deo, 2016; Shen et al., 2022). The task of graph generation aims to create new graph structures that satisfy specific properties, holding significant scientific and practical value for key scenarios like social network analysis (Wasserman, 1994), molecular discovery (Elton et al., 2019), the optimization of transportation and communication networks (Jiang, 2022), *etc.*

Over the past decade, deep learning models represented by Graph Neural Networks (GNNs) (Wu et al., 2020; Kipf, 2016; Hamilton et al., 2017; Veličković et al., 2018; Xu et al., 2019), Variational Autoencoders (VAEs) (Kingma & Welling, 2013), and Generative Adversarial Networks (GANs) (Goodfellow et al., 2014) have achieved remarkable success in the field of graph generation (Zhu et al., 2022; You et al., 2018; Simonovsky & Komodakis, 2018; Wang et al., 2018). However, these traditional graph generation methods are typically trained on domain-specific data, exhibit weak generalization capabilities, and cannot comprehend complex instructions given in natural language, making them difficult to adapt to the new paradigm of *Text-Instructed Graph Generation* centered around language requirements. These models typically take graph structures (*e.g.*, GNN-based models) or latent vectors (*e.g.*, VAEs and GANs) as input, as shown in Figure 1(a). Recently, benefiting from the outstanding performance of Large Language Models (LLMs) (Zhao et al., 2023; Mann et al., 2020; Touvron et al., 2023) in generalization and semantic understanding, researchers have begun to explore the paradigm of *Text-Instructed Graph Generation*. This paradigm

**(a) Deep Learning–based Graph Generation Paradigm**

Input Vector $\mathbf{v}_i$

Input Graph

DNN Model

Adjacency matrix

Vector matrix

Generated Graph

Generate a new acyclic compound

LLM

User

Here's a hypothetical acyclic compound: 2-hydroxy-3-pentanone

Parse

Generated Graph

**(b) Text-Instructed Graph Generation Paradigm in LLM-Era**

Figure 1: Paradigm Shift in Graph Generation: Traditional DeepLearning *vs.* LLMEra Conversational TextInstructed Graph Generation

enables models to directly generate corresponding graph structures based on users' natural language instructions (Yao et al., 2024; Li et al., 2024), as illustrated in Figure 1(b).

Despite the exciting prospects of text-instructed graph generation, the development of this field is facing a critical bottleneck: *a severe lack of benchmarks specifically designed for this new paradigm*. This deficiency prevents a reliable evaluation and in-depth analysis of the true capabilities of existing models. The current predicament is mainly manifested in two aspects.

First, there is a lack of high-quality datasets tailored for the text-instructed graph generation task. Few existing evaluation efforts (Wang et al., 2023; Huang et al., 2023) often rely on simple synthetic tasks, (*e.g.*, "generate a cycle graph with 5 nodes", or "construct an undirected acyclic graph"), falling far short of reflecting the complexity of real-world applications. Besides, while traditional graph generation benchmarks (Ramakrishnan et al., 2014b; Gilmer et al., 2017; Hu et al., 2020b)) contain rich graph data, they do not include diverse natural language instructions paired with the graph structures, failing to meet the requirements of the text-instructed graph generation task. Second, the evaluation systems are inadequate and fail to comprehensively measure model performance. Current evaluations of graph generation largely inherit the mindset of traditional methods, judging the "likeness" by comparing the similarity of macroscopic statistical features (Li et al., 2018; Stokes et al., 2020) between the generated graph and a reference graph (You et al., 2018; O'Bray et al., 2021). However, text-instructed generation requires a deeper level of evaluation, specifically: (1) Semantic Fidelity, the generated graph must faithfully adhere to every explicit or implicit constraint provided in the text instruction; and (2) Domain Knowledge Alignment, the generated graph must conform to the inherent rules of its domain—knowledge that is not explicitly stated in the text.

To fill the aforementioned gaps, we propose Text2GraphBench[1], a comprehensive benchmark specifically designed for evaluating and analyzing the performance of generative models on text-instructed graph generation tasks.

We pioneer a "*Graph-to-Constraints-to-Text*" generation pipeline. First, we collect raw graph data from multiple domains and then automatically extract a verifiable Constraint Set for each graph. This set includes not only explicit structural attributes like node count and diameter but also implicit domain knowledge such as chemical bonding rules and small-world properties. Then, using this constraint set as the ground truth, we render text instructions in various forms, including natural language and formal specifications, ensuring that the requirements of each instruction are precisely verifiable (Lin et al., 2023; Osuji et al., 2024; Puduppully et al., 2019). Additionally, we propose an evaluation paradigm based on the Constraint Pass Rate. For any generated graph, we directly use its corresponding constraint set as a test suite, automatically checking the pass rate of each constraint. This framework quantifies the model's ability to follow explicit instructions and examines its grasp of implicit domain rules.

Our contributions are three-fold. **(1) A Pioneering Benchmark for Text-instructed Graph Generation.** We construct and release a large-scale, multi-domain dataset. It covers a full spectrum of topological structures, from scale-free social networks to grid-like transportation meshes. Beyond a

---

[1]Anonymous link: `https://anonymous.4open.science/r/Text2GraphBench-68C7`.

dataset, we provide a unified, extensible pipeline that serves as a bridge between the graph and LLM communities, enabling verifiable evaluation of structured reasoning capabilities.

**(2) A Three-Dimensional Evaluation Framework Centered on Constraints.** We move beyond traditional, ambiguous evaluations based on statistical similarity and propose a novel, multi-dimensional evaluation system centered on constraints. This system decomposes model performance into three core aspects: Validity, Semantic Faithfulness, and Novelty & Diversity. **(3) Extensive Evaluation and In-depth Analysis of Existing Large Models.** We conduct systematic experiments on a series of current mainstream LLMs. The results provide the first systematic revelation of their current capabilities, common strengths, and shared challenges in text-instructed graph generation, offering key insights for future improvements.

## 2 TEXT2GRAPHBENCH BENCHMARK

### 2.1 DESIGN PRINCIPLES

The design of Text2GraphBench benchmark is centered around four core objectives: verifiability, diversity, extensibility, and reproducibility. It adheres to the following three foundational principles.

**Verifiability:** This is our foremost design principle. We propose a "constraint-centric" paradigm, wherein each natural language instruction is intrinsically linked to a set of precise, quantifiable constraints that serve as the ground truth for evaluation. This design enables the fully automated assessment of model-generated outputs, fundamentally ensuring the objectivity and reproducibility of the results.

**Diversity and Graded Complexity:** To comprehensively characterize model capabilities, our benchmark covers not only abstract synthetic graphs but also four specific domain-specific graph types with real-world application value. Within each domain, we have designed a task ladder ranging from simple (*e.g.*, following a single, explicit structural constraint) to complex (*e.g.*, requiring the understanding of multiple combined constraints and implicit domain knowledge), aiming to probe the capability boundaries of models in a fine-grained manner.

**Extensibility and Usability:** We employ a standardized data format (*i.e.*, JSON[2]) and provide a complete Python toolchain that supports data loading, constraint computation, and automated evaluation. This modular design allows researchers to easily extend our paradigm to new domains or add new constraint types within existing ones, thereby robustly promoting open collaboration and future research within the community.

### 2.2 THE CONSTRUCTION OF TEXT2GRAPHBENCH

The construction of Text2GraphBench strictly follows the "constraint-centric" paradigm outlined in Section 2.1. The core idea is to use machine-verifiable constraints as a bridge to connect raw graph data with the final natural language instructions. This "Graph-to-Constraint-to-Text" pipeline not only ensures that the semantics of the instructions are rigorously aligned with the graph structures but also lays a solid foundation for subsequent automated evaluation. Specifically, our construction process, as illustrated in Figure 2, consists of the following three key steps.

#### 2.2.1 RAW GRAPH DATA ACQUISITION AND PREPROCESSING

We begin by collecting six public datasets from four representative real-world domains to ensure the benchmark's diversity and utility as shown in Figure 2(a). These domains include:

- **Molecular Graphs**: QM9 (Ramakrishnan et al., 2014b)[3], ZINC (Irwin & Shoichet, 2005)[4];
- **Social Networks**: DBLP (Ley, 2002)[5], Reddit (Leskovec & Sosič, 2016)[6];

---

[2]https://www.json.org/json-en.html.
[3]https://www.nature.com/articles/sdata201422.
[4]https://zinc.docking.org/.
[5]https://dblp.org/.
[6]https://github.com/Zhuofeng-Li/TEG-Benchmark.

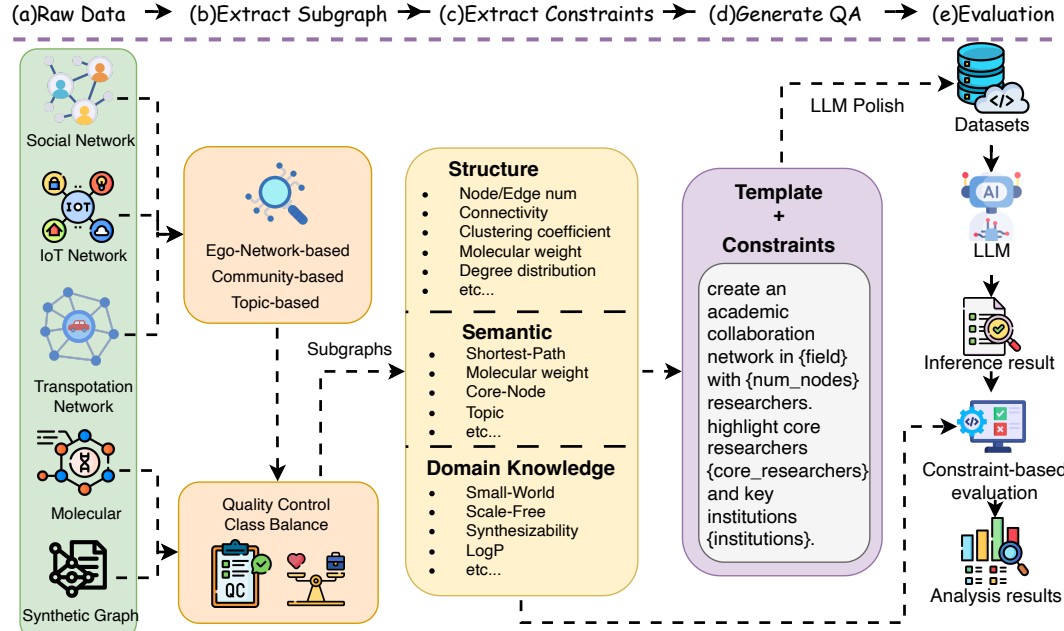

Figure 2: The construction pipeline of Text2GraphBench.

- **Transportation Networks**: `PEMS-BAY` (Li et al., 2017)[7];
- **Internet of Things (IoT)**: `IoT Intrusion Dataset` (Ullah & Mahmoud, 2020)[8].

Based on the intrinsic structure of these datasets, we adopt two distinct preprocessing strategies as shown in Figure 2(b). (1) For datasets composed of a large number of small, independent graphs (*e.g.*, `QM9` and `ZINC`), each molecule naturally serves as an individual graph sample. We directly perform quality filtering on these samples, removing those that are overly simple or anomalous, followed by stratified sampling to construct a difficulty-balanced test set. (2) For datasets consisting of a single large-scale network (*e.g.*, `DBLP`, `PEMS-BAY`), We utilize the **Louvain algorithm** to extract research communities from DBLP and **k-hop BFS sampling** for PEMS-BAY's geographic topology.

Crucially, to mitigate sampling bias and prevent high-degree core inflation, we implement the uniform random seed strategy across the entire graph. This ensures a balanced representation of structural densities, ranging from sparse peripheries to dense cores. Detailed sampling specifications are provided in Appendix B.2. All preprocessed and sampled graphs then proceed to the next stage of constraint extraction.

### 2.2.2 CONSTRAINT EXTRACTION AND STRATIFICATION

This step is crucial for bridging graph structures with natural language instructions. To ensure the validity of constraints and the hierarchical nature of the evaluation, we collaborated with domain experts to design and implement a three-tiered constraint system.

**(1) Structural Constraints**: These describe the macroscopic topological properties of a graph and are designed to test a model's understanding of fundamental graph theory concepts, such as the exact number of nodes and edges, graph density, diameter, and average degree.

**(2) Domain-Specific Constraints**: These are tightly coupled with the specialized knowledge of a particular domain, requiring the model to comprehend semantics beyond pure topology. For exam-

---

[7]https://www.kaggle.com/datasets/scchuy/pemsbay.
[8]https://mivia.unisa.it/datasets/iot-network-analysis/.

ple, in molecular graphs, this could refer to "containing at least one carbonyl group (C=O)" or the number of aromatic rings.

**(3) Domain Knowledge Constraints**: These represent the implicit "physical laws" or "common sense" rules of a domain that users typically do not specify in their instructions. For instance, a chemically valid molecule must obey valency rules. This class of constraints is used exclusively during evaluation to assess the realism and plausibility of the generated graph, without being included in the model's input instructions, thus preventing information leakage.

During the extraction phase, we implement precise computation functions for each constraint type. For any given graph sample, we automatically calculate its statistical values across these constraint dimensions and store them as a structured, machine-readable list, providing the "ground truth" for subsequent instruction generation and automated evaluation.

### 2.2.3 CONSTRAINT-DRIVEN INSTRUCTION GENERATION

The final step aims to convert the structured list of constraints into fluent, diverse, and natural-sounding language instructions. To balance accuracy and naturalness, we designed a two-stage generation process.

**(1) Template-Based Initial Generation**: We first randomly select a combination of 1-3 constraints from the structural and domain-specific categories for each graph sample. We then programmatically populate these constraint values into a series of predefined, domain-specific text templates to generate an initial instruction. To ensure the logical feasibility of the tasks, we strictly sample constraint combinations extracted from the same source graph. This "existence proof" guarantees that all constraints within a single instruction are mutually compatible by design, effectively eliminating the risk of generating contradictory or impossible requirements.

**(2) LLM-Based Language Polishing**: Since template-generated text can be mechanical and repetitive, we pass the initial instruction to an LLM (we use models from the Qwen series) for language "polishing". The model reframes the template-based input into a more natural and diverse expression while ensuring that the key information from the original constraints is preserved.

**(3) Automated Consistency Verification**: To prevent semantic drift during LLM polishing (e.g., hallucinated numerical values or operator flipping), we integrate a strict post-validation module. We employ a rule-based reverse parser to extract constraints back from the generated instruction and perform an exact-match check against the ground truth. Any instruction failing this bidirectional consistency verification is automatically discarded and regenerated, ensuring 100% alignment between text and constraints.

This automated pipeline enables us to efficiently construct a large-scale instruction dataset where each instruction is strictly tied to one or more verifiable constraints, while remaining rich and varied in its linguistic expression.

A typical data sample is formatted as shown in Listing **??**, which clearly illustrates the relationship between the instruction, constraints, and metadata.

### 2.3 DATASET COMPOSITION

To ensure the diversity and comprehensiveness of our benchmark, we apply the aforementioned data construction paradigm to two major categories, "General Graphs" and "Domain Graphs", to build the core dataset of Text2GraphBench.

**General Graphs** This part aims to evaluate a model's fundamental understanding of abstract graph theory concepts, without reliance on any specialized domain knowledge. We utilize NetworkX[9], a mature graph computation library, to programmatically generate a series of classic graph structures. We select Erdős–Rényi (ER) random graphs (Erdős et al., 1960), Watts–Strogatz (WS) small-world graphs (Watts & Strogatz, 1998), and Barabási–Albert (BA) scale-free graphs (Barabási et al., 1999), since they represent the three most fundamental and core network formation mechanisms in network science: "complete randomness", "small-world properties", and "growth and preferential attach-

---

[9] https://networkx.org/.

---

**Data Sample Example**

```
1  {
2    "id": "mol_zinc_12345",
3    "domain": "molecular graph",
4    "instruction": "please generate a molecule, it needs to contain a
          6-membered ring structure...",
5    "constraints": [
6      {
7        "structural": {
8            "type": "node_count",
9            "operator": "between",
10           "value": [15, 20]
11       },
12       "semantic": {
13           "type": "atom_type_count",
14           "atom": "O",
15           "operator": ">=",
16           "value": 3
17       }
18     }
19   ],
20   "meta_data": {
21     "graph_type": "molecule",
22     "source_data": "ZINC",
23     "difficulty": "hard"
24   }
25  }
```

Figure 3: An example of a data sample in Text2GraphBench.

ment", respectively. They serve as a cornerstone for evaluating a model's general graph reasoning abilities and are widely adopted in existing graph generation work. The instructions for these tasks directly describe the graph-theoretic concepts, *e.g.*, "Generate an ER random graph with 15 nodes and 20 edges".

**Domain Graphs**   This part aims to evaluate a model's performance in real-world scenarios that require deep domain knowledge. We have selected four domains with broad representativeness and significant application value, constructing data subsets for each as follows.

• **Molecular Graphs**: Based on the `QM9` and `ZINC` datasets, where graphs represent atoms and chemical bonds. Instructions specify structural and semantic constraints such as atom/bond counts, ring structures, and functional groups. The evaluation also implicitly verifies adherence to chemical valency rules and assesses properties like quantitative estimation of drug-likeness (QED) and synthetic accessibility (SA) scores.

• **Social Networks**: Based on the `DBLP` and `Reddit` datasets, where graphs represent users/authors and their relationships. Instructions specify constraints like the number of communities, network diameter, and core node degrees. The evaluation also checks whether the generated network exhibits typical topological properties of real social networks, such as the "small-world" effect and "scale-free" degree distributions.

• **Transportation Networks**: Based on the `PEMS-BAY` dataset, where graphs represent sensors/intersections and road segments. Instructions specify constraints such as node scale, arterial road connectivity, and intersection density. The evaluation also assesses the global reachability of the generated road network (to avoid isolated "island" regions) and the rationality of its topological hierarchy.

• **IoT Security**: Based on the `IoT Intrusion Dataset`, where graphs represent communication relationships between devices. Instructions specify constraints like device type distribution

and protocol usage patterns. The evaluation also verifies whether the generated graph complies with logical protocol interaction rules for specific scenarios, for instance, detecting anomalous direct connections between two end devices that should communicate via a gateway.

## 2.4 DATASET STATISTICS OVERVIEW

To provide an intuitive overview of the scale and characteristics of Text2GraphBench, we conduct a detailed statistical analysis of the entire dataset. Table 1 summarizes the key statistical features of the core domain datasets.

Table 1: Statistical overview of the Text2GraphBench dataset. This table details the data source, number of samples, difficulty distribution (Easy/Medium/Hard), average number of constraints, and key domain-specific constraints for each domain.

| Domain | Data Source | #(Samples) | Difficulty (E/M/H) | Avg. Constraints | Key Domain Constraints |
|---|---|---|---|---|---|
| Synthetic Graphs | Erdos-Renyi Graph | 10,000 | 3,300 / 3,400 / 3,300 | 3.4 | Node count, Edge probability, Directed, Connectivity |
| | Barabasi-Albert Graph | 9,334 | 3,187 / 3,082 / 3,082 | 2.0 | Node count, Attachment edges |
| | Watts-Strogatz Graph | 10,000 | 3,300 / 3,400 / 3,300 | 4.4 | Node count, Directed, Connectivity |
| | Complete | 9,334 | 3,300 / 3,400 / 3,300 | 4.0 | Node count, Edge count, Density, Connectivity |
| Molecular Graphs | QM9 | 9,939 | – / 7,544 / 2,395 | 5.2 | Atom/bond counts, Ring structures, Aromatic Ring Count |
| | ZINC | 10,000 | – / 7,436 / 2,564 | 5.3 | Element Ratio C, Bond type count single |
| Social Networks | DBLP | 7,281 | – / 3,641 / 3,640 | 6.0 | Six-degrees of separation, Small-world properties, Scale-free exponent, Key entities |
| | Reddit | 8,000 | – / 3,984 / 4,016 | 6.5 | Codularity, Topic distribution, Node type distribution, Anomalous communication |
| Transportation Networks | PEMS-BAY | 10,000 | – / 5,104 / 4,896 | 5.6 | Network scale, Connectivity, Arterial road count, Global reachability |
| IoT Security | IoT Intrusion | 10,000 | – / 5,024 / 4,976 | 5.6 | Device types, Protocol distribution, Traffic patterns, Anomalous communication |
| **Total** | | **94,554** | **12,965 / 46,120 / 35,469** | **4.66** | – |

## 2.5 EVALUATION FRAMEWORK

To ensure a rigorous and comprehensive assessment, we have designed a three-dimensional evaluation framework that is tightly integrated with our constraint-centric data construction paradigm. Before any evaluation, a model's output must pass a basic parsing check (i.e., it can be successfully parsed into a valid graph object); any unparsable output is immediately marked as a generation failure. For successfully parsed graphs, we assess them along the following three progressive dimensions:

**Structural Fidelity**: This dimension evaluates whether the generated graph accurately adheres to all explicit, quantifiable **structural constraints** specified in the instruction. It directly tests the model's low-level instruction-following capabilities by focusing on the accuracy of the graph's topological properties.

**Semantic Accuracy**: Building on structural correctness, this dimension assesses whether the generated graph conforms to the **domain-specific concepts** described in the instruction. This requires the model to understand not just numbers but also the topological mapping of concepts. For example, an instruction for "a graph representing a cyclopropane molecule" requires generating a triangle, not just a 3-node path.

**Domain Plausibility**: This dimension evaluates whether the generated graph adheres to the **implicit, unstated intrinsic rules** of its domain. This is the highest level of evaluation, measuring whether

the model has acquired the domain knowledge needed to generate "plausible" graphs. For example, does a generated molecule obey chemical valency rules?

This three-dimensional framework ensures a layered evaluation: first, we check if the model can "listen" to instructions, then if it can "understand" concepts, and finally, if it has "mastered" the knowledge. The verifiable constraints produced during data construction serve directly as the evaluation criteria, creating a natural closed loop and ensuring consistency and reproducibility across all domains and difficulty levels.

### 2.5.1 EVALUATION PROCESS AND METRICS

Our evaluation process is designed as a fully automated pipeline to ensure efficiency and objectivity. The steps are as follows:

**Model Inference**: We input all instructions from the TEXT2GRAPHBENCH dataset into the LLM being evaluated and collect the text-formatted outputs.

**Output Parsing**: We employ a robust parser that attempts to convert the model's text output into a standard NetworkX graph object using multiple strategies (e.g., adjacency lists, GML, Graphviz DOT). If all parsing strategies fail, the sample is marked as a failure.

**Constraint Validation**: For each successfully parsed graph, a validation module retrieves the corresponding list of constraints and invokes independently written validation functions for each constraint to perform checks. To ensure reproducibility, we strictly adhere to standard mathematical definitions and statistical thresholds for complex structural metrics. The precise formulations and implementation details are provided in Appendix B.2.

**Metric Calculation**: Finally, we compute our core quantitative metric based on the validation results. We define the Constraint Pass Rate (CPR) as our primary evaluation metric, which is the percentage of total constraints successfully met by the model across all test samples. We also report CPR scores for each of the three constraint dimensions (structural, semantic, and domain) to enable a more fine-grained analysis. A high CPR value indicates that a model possesses a strong and precise capability for generating graphs that follow complex instructions.

## 3 EXPERIMENTS

### 3.1 EVALUATED MODELS

We have carefully selected a series of representative models with broad influence in the current AI landscape, aiming to cover diverse model architectures, parameter scales, and technical routes. The lineup of evaluated models is divided into two main categories.

**(1) Closed-Source Models:** We selected two industry-leading closed-source models as performance benchmarks: Qwen-Plus (Yang et al., 2025) and Gemini-2.5-Flash (Comanici et al., 2025).

**(2) Open-Source Models:** We chose several mainstream open-source model families that have garnered significant attention in the community, including DeepSeek-V3 (Liu et al., 2024), Llama-3-70B-Instruct (Grattafiori et al., 2024), Qwen-32B (Yang et al., 2025), GPT-OSS-20B (Agarwal et al., 2025), and Llama4-Scout (Grattafiori et al., 2024).

We provide more details of experimental setups in Appendix E.

### 3.2 OVERALL PERFORMANCE OVERVIEW

We evaluate the comprehensive performance of all models on the entire test set of Text2GraphBench. Table 2 summarizes the CPR of each model in five main domains and overall.

Based on the results from Table 2, we can draw several key preliminary conclusions:

1. *A Clear Hierarchy in Model Capabilities, with Closed-Source Models Excelling:* "gemini-2.5-flash" demonstrates the strongest overall capability among all evaluated models, achieving a total CPR of 65.1%. It secures the top score across all 10 sub-datasets, significantly outperforming other

Table 2: Model Performance on Synthetic and Real-world Graph Datasets (CPR %)

| Model | Synthetic Graphs | | | | Molecular | | Social | | Transport | IoT | Overall |
|---|---|---|---|---|---|---|---|---|---|---|---|
| | ER | BA | WS | Complete | QM9 | ZINC | DBLP | Reddit | PEMS | IoT | Avg. |
| gemini-2.5-flash | 62.7 | 40.6 | 20.9 | 42.3 | 72.3 | 69.8 | 75.2 | 73.9 | 68.4 | 70.1 | **65.1** |
| llama-v3-70b-instruct | 39.4 | 3.2 | 24.4 | 40.3 | 58.4 | 56.1 | 61.5 | 60.2 | 55.3 | 57.8 | **51.6** |
| deepseek-v3 | 43.6 | 20.3 | 9.2 | 13.2 | 55.2 | 53.7 | 58.9 | 57.4 | 52.1 | 54.6 | **48.2** |
| qwen-plus | 42.9 | 19.6 | 9.8 | 14.0 | 54.8 | 53.2 | 58.3 | 56.9 | 51.7 | 54.1 | **47.8** |
| llama4-scout | 38.4 | 1.3 | 30.7 | 37.2 | 52.1 | 50.3 | 56.8 | 55.1 | 48.9 | 51.2 | **46.8** |
| gpt-oss-20b | 40.3 | 0.8 | 24.5 | 42.0 | 43.8 | 41.2 | 46.3 | 45.1 | 40.6 | 42.9 | **39.7** |
| qwen3-32b | 39.5 | 0.9 | 26.1 | 23.8 | 25.4 | 23.1 | 28.7 | 27.3 | 22.8 | 24.5 | **24.4** |
| **Avg. by Dataset** | 43.8 | 12.4 | 20.8 | 30.4 | 51.7 | 49.6 | 55.1 | 53.7 | 48.5 | 50.7 | |

models. This indicates its comprehensive superiority in adhering to complex instructions and generating precise graph structures.

2. *Significant Domain-Specific Performance Disparities:* The models' performance varies dramatically across different domains. For instance, all models generally achieve a higher CPR on real-world graphs rich in semantic information, such as molecular graphs (QM9, ZINC) and social networks (DBLP, Reddit), compared to abstract synthetic graphs. Notably, on the most challenging Barabási–Albert (BA) graphs, the average CPR across all models is a mere 12.4%, highlighting that generating scale-free networks with specific degree distributions is a common and formidable challenge for current LLMs.

3. *The Inherent Difficulty of the Task:* Even the top-performing model, "gemini-2.5-flash", is far from achieving an ideal overall pass rate. This underscores the inherent high difficulty of the text-to-graph generation task, which places extreme demands on a model's logical reasoning, spatial conceptualization, and domain knowledge integration capabilities.

4. *A Non-Linear Relationship Between Performance and Parameter Scale:* Across different model families, a larger parameter count does not always guarantee better performance. For example, while the 70B-parameter "llama-v3-70b-instruct" is the top performer within the Llama family, its overall performance (51.6%) is still surpassed by "gemini-2.5-flash". This suggests that differences in model architecture, training data, and alignment strategies play a more critical role than parameter scale alone in determining graph generation capabilities.

5. *Root Causes of Performance Disparities.* Beyond these general trends, our in-depth attribution analysis reveals that performance gaps arise from systematic limitations rather than random noise. A primary bottleneck for lower-performing models is *format instability*, where models like Qwen3-32B struggle to maintain syntax due to their 'Thinking Mode' drifting into unstructured reasoning compared to Gemini's near-perfect adherence. Furthermore, we observe a sharp divergence between "semantic generation" and "mathematical simulation." While models leverage pre-trained knowledge for semantic-rich domains, they suffer from a *simulation deficit* on synthetic tasks, lacking the internal engine to execute dynamic mathematical rules. Finally, performance degrades significantly as constraint counts exceed 5, indicating that autoregressive architectures inherently struggle to maintain global topological consistency under high reasoning loads.

### 3.3 OTHER DISCUSSIONS

Due to space limitations, we provide more experimental discussions in Appendix G.

## 4 RELATED WORK

**Traditional Graph Generation Models.** Traditional graph generation models aim to learn the probability distribution $p(G)$ of graphs to generate structures with similar properties. Prominent approaches include: (1) Auto-regressive models (You et al., 2018), which generate graphs through a sequence of decisions (e.g., adding nodes and edges one by one) but suffer from order sensitivity and error accumulation; (2) Generative Adversarial Networks (GANs) (Wang et al., 2018), which use adversarial training between a generator and a discriminator to learn complex graph patterns but face challenges like training instability and mode collapse; and (3) Variational Autoencoders

(VAEs) (Simonovsky & Komodakis, 2018), which generate graphs by sampling from a latent space. More recently, diffusion models (Chamberlain et al., 2021) have demonstrated significant potential in generation quality. However, these traditional models are fundamentally unconditional or accept only simple conditions. Their core objective is to replicate an existing structural distribution rather than to understand and execute complex natural-language instructions.

**Graph Task Benchmarks.** The development of graph task benchmarks has significantly advanced the field of graph machine learning. Benchmarks such as OGB (Hu et al., 2020a) primarily serve discriminative tasks like node classification, evaluating a model's ability to understand existing graph structures. For traditional unconditional generation, the community has also established domain-specific benchmarks, such as QM9 (Ramakrishnan et al., 2014a) for molecular generation and various community network datasets for social network analysis, where evaluation focuses on comparing the statistical similarity between generated and real graphs. In summary, neither benchmarks for discriminative tasks nor those for unconditional generation can meet the evaluation needs of text-instructed graph generation models.

 **LLMs on Graph Tasks.** The rapid progress of LLMs has sparked growing interest in applying them to graph-related problems. *LLMs for graph analysis and reasoning.* A common approach linearizes graph structures into text for tasks such as node classification or link prediction (Jin et al., 2024; Li et al., 2023). Beyond this paradigm, GraphLLM (Chai et al., 2023) integrates GNNs with LLMs to alleviate the information loss of Graph2Text. GraphInstruct (Luo et al., 2024) provides a 21-task reasoning benchmark with instruction-tuned models. InstructGraph (Wang et al., 2024) unifies graph representation through a structured verbalizer and applies graph-centric instruction tuning. These efforts primarily enhance LLMs' ability to *understand* graph structures.

*LLMs for graph generation and graph–text modeling.* Early explorations prompt LLMs to output edge lists or adjacency matrices (Yao et al., 2024). More recent work evaluates LLMs' graph-generation capability through code-based structural optimization (Wang et al., 2025). Davies (Davies, 2025) further highlights open challenges and frontiers in graph generation. On the graph–text side, PlanGTG (He et al., 2025) examines graph-to-text generation and improves planning and grounding via reordering and attribution tasks.

In summary, existing studies demonstrate the promise of LLMs for graph understanding and generation, but remain task-specific or exploratory. To our knowledge, no prior work offers a standardized, large-scale, constraint-centric benchmark for *text-instructed graph generation* across diverse domains—this is precisely the contribution of Text2GraphBench.

## 5 CONCLUSIONS

In this work, we address the lack of standardized evaluation for LLMs on text-instructed graph generation by introducing the first comprehensive benchmark, Text2GraphBench. We first clarify that current evaluation practices are bottlenecked by the absence of high-quality text–graph paired datasets and by the lack of metrics that can assess semantic faithfulness. To remedy this, we present Text2GraphBench, whose core is a constraint-centered methodology for both dataset curation and evaluation. On the dataset side, we implement a "graph-to-constraint, constraint-to-text" pipeline to construct a large-scale, multi-domain corpus that guarantees the verifiability of each instruction. On the evaluation side, we move beyond traditional similarity-based comparisons by proposing a three-dimensional framework that assesses Validity, Semantic Fidelity, and Novelty and Diversity. Extensive experiments on Text2GraphBench provide the first systematic characterization of mainstream models' capabilities on this task, including their common strengths and persistent shortcomings. Our results indicate that, despite substantial promise, a significant gap remains between general language understanding and precise, domain-compliant graph generation.

# 6 ETHICS STATEMENT

We strictly adhere to the ICLR Code of Ethics. The core objective of this research is to construct a comprehensive and rigorous benchmark, Text2GraphBench, for evaluating and understanding the capabilities of Large Language Models on text-instructed graph generation tasks. Our aim is to foster more transparent and credible research in this field. We confirm the following:

1. **Data Sources and Privacy:** All raw graph data used to build Text2GraphBench originate from public, anonymized academic and public datasets (e.g., DBLP, QM9). Our data generation and processing pipeline does not involve any personally identifiable information or private user data.

2. **Potential Bias and Fairness:** We have made a concerted effort to ensure our dataset covers multiple diverse domains (e.g., general graphs, molecular graphs, knowledge graphs) to mitigate biases that could arise from a single domain focus. The text instructions we generate are designed to objectively describe graph properties and do not contain any discriminatory or offensive language.

3. **Research Purpose and Potential Impact:** This research is intended to provide an evaluation tool with the direct purpose of advancing technology. We recognize that any powerful generative model carries a potential risk of misuse. We encourage the community to be mindful of and guard against the possibility of generating harmful or misleading graph information when using Text2GraphBench and associated models. Our open-source license will explicitly prohibit the use of Text2GraphBench for malicious purposes.

We believe this work is a positive and beneficial contribution to the community, facilitating a deeper understanding and improvement of LLMs.

# 7 REPRODUCIBILITY STATEMENT

To ensure the full reproducibility of our research, we have made the following efforts and commit to releasing all relevant code and data upon acceptance of the paper:

1. **Dataset and Code:** The complete Text2GraphBench dataset, including graph data from all domains, the generated constraint sets, and the text instructions, will be released via the Hugging Face Datasets platform. All source code for data generation, model evaluation, and results analysis will be hosted in a public, anonymous GitHub repository. Detailed dataset statistics and the generation pipeline are described in Section 2.

2. **Experimental Setup:** We provide a detailed description of all experimental setups in Section 3, including the baseline models used, hyperparameter configurations, and the specific calculation methods for our evaluation metrics, to ensure that our results can be precisely reproduced.

3. **Computational Environment:** We document the hardware environment (e.g., GPU models) and key software dependencies (e.g., PyTorch, NetworkX) with their version numbers used to run the experiments.

We are committed to providing an open, easy-to-use, and extensible benchmarking ecosystem to foster sustainable and reproducible research in this field.

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

APPENDIX

## A  USE OF LLMS

In the preparation of this manuscript, we utilized LLMs for two primary purposes: code generation and english language polishing. The core scientific ideas, experimental design, theoretical contributions, and the overall narrative of the paper were conceived and written entirely by the human authors. LLMs served as a tool to assist in the implementation of certain code segments and to improve the clarity, grammar, and readability of the text.

## B  TEXT2GRAPHBENCH BENCHMARK DETAILS

### B.1  DETAILED SUBGRAPH SAMPLING SPECIFICATIONS

For the social network domain (DBLP), we employ the Louvain community detection algorithm to identify dense research groups. We set the resolution parameter to 1.0 to capture medium-sized communities and sample cluster sizes from a clipped normal distribution to ensure scale diversity. For the transportation network (PEMS-BAY), we utilize a k-hop Breadth-First Search (BFS) approach, where each step expands to a maximum of 5 neighbors to preserve local geographic topology. Most critically, to mitigate potential selection biases such as high-degree core inflation, we adopt a "Uniform Random Seed" strategy for both domains. By initiating sampling from uniformly distributed nodes across the entire graph rather than high-centrality nodes, we ensure that our dataset captures a diverse spectrum of structural features, covering both dense cores and sparse peripheries.

### B.2  PRECISE DEFINITIONS OF STRUCTURAL METRICS

To ensure the auditability and reproducibility of our evaluation, we provide the exact mathematical formulations and thresholds for the structural metrics used in Text2GraphBench.

1. *Small-World Coefficient ($\sigma$):* We adopt the Watts–Strogatz definition.

$$\sigma = \frac{C/C_{\text{rand}}}{L/L_{\text{rand}}} \tag{1}$$

where $C$ and $L$ are the clustering coefficient and average shortest path length of the generated graph, respectively. $C_{\text{rand}}$ and $L_{\text{rand}}$ are the corresponding values of an Erdős–Rényi random graph with equivalent size and density. Following Humphries & Gurney, we classify a graph as small-world if $\sigma > 3$ and $C > 0.1$.

2. *Scale-Free Fit (Power-Law):* We employ the rigorous Clauset–Shalizi–Newman (CSN) statistical framework via the `powerlaw` package. We estimate the exponent $\alpha$ and $x_{\min}$ using Maximum Likelihood Estimation (MLE) and assess goodness-of-fit via the Kolmogorov–Smirnov (KS) statistic. A graph is considered scale-free if: $1.5 \leq \alpha \leq 4.0$ and $p_{\text{KS}} > 0.1$ where $p_{\text{KS}} > 0.1$ indicates that the power-law hypothesis cannot be rejected.

3. *Assortativity ($r$):* We calculate Newman's assortativity coefficient $r$. In our evaluation, we allow for reasonable structural variability by defining the acceptance threshold as an interval: $[r_{\text{true}} - 0.2, r_{\text{true}} + 0.2]$, where $r_{\text{true}}$ is the value extracted from the source graph.

## C  DESIGN PHILOSOPHY AND INTENDED USE

The core design philosophy of Text2GraphBench is to be **constraint-centric and verifiable**. We aim to address the current lack of standardized evaluation in the field by providing researchers with a platform to accurately measure the ability of Large Language Models (LLMs) and Graph Foundation Models (GFMs) to generate graph structures following complex natural language instructions.

This benchmark is primarily intended for the following research and development communities:

- **Large Language Model Developers:** To evaluate and enhance their models' capabilities in logical reasoning and precise control over structured data generation, particularly for graph data.

- **Graph Machine Learning Researchers:** To explore the potential and bottlenecks of using LLMs as knowledge sources or "world models" for graph learning tasks such as graph generation, editing, and reasoning.

- **Cross-Disciplinary Application Explorers:** To investigate the application potential of LLMs in domains like automated scientific discovery (e.g., molecular design), software engineering (e.g., code structure generation), and cybersecurity (e.g., attack graph simulation) using this benchmark.

## C.1 HOSTING, MAINTENANCE, AND VERSION CONTROL

To ensure the long-term availability and reproducibility of the benchmark, all components of Text2GraphBench—including the dataset, evaluation code, and model outputs—will be hosted and maintained as follows:

- **Code and Dataset Repository:** Hosted on GitHub and iterated through version control. The link for anonymous review is: `https://anonymous.4open.science/r/Text2GraphBench-68C7`.

- **Dataset Distribution:** Convenient download and usage will be provided via the Hugging Face Datasets Hub.

- **Maintenance Commitment:** The core team commits to actively maintaining this project for at least the next two years, which includes fixing bugs, answering community questions, and periodically updating the benchmark (e.g., by incorporating new models, task domains, or advanced evaluation metrics) in line with developments in the field.

## C.2 DISCUSSION OF LIMITATIONS

Although Text2GraphBench represents a significant effort in comprehensiveness and rigor, as an early exploration in this direction, it still has the following limitations:

- **Limited Domain Coverage:** While the current five domains are representative, they are far from covering all important graph application scenarios.

- **Simplified Instruction Expression:** The constraint-based method, while ensuring verifiability, may not fully capture certain vague, implicit, or ambiguous nuances of real-world human instructions.

- **Incomplete Evaluation Dimensions:** The current evaluation framework primarily focuses on the correctness and diversity of the generated results, with insufficient consideration for dimensions such as generation efficiency and interpretability.

We view these limitations as directions for future work and welcome the community to collaborate with us in expanding and refining Text2GraphBench.

## D COMPLETE CONSTRAINT TAXONOMY

The evaluation framework of Text2GraphBench is built around three major categories of constraints. The table below (Table 3) provides a complete taxonomy and its instantiation across different domains.

## E EXPERIMENTAL SETUPS

**Implementation Details** All model inferences were conducted via their official APIs or executed on a local server using the vLLM (v0.8.5) (Kwon et al., 2023) framework. The local experimental environment was configured with a 16-core CPU, 96GB of RAM, and an NVIDIA RTX 3090 (24GB)

Table 3: The Complete Constraint Taxonomy. This table details the three main categories of constraints—Structural, Semantic, and Domain-specific—along with their subcategories and concrete examples.

| Constraint Category | Subcategory / Domain | Example Constraints |
|---|---|---|
| **Structural** | Global Topology | Number of nodes/edges, graph density, connectivity, graph diameter, average path length, bipartiteness. |
| | Local Topology | Maximum/minimum/average node degree, count of cycles of a specific length (e.g., triangles), presence/absence of specific motifs (e.g., stars, cliques), node degree distribution. |
| **Semantic** | Node/Edge Attributes | Type and label of nodes/edges, count of nodes/edges of a specific type, numerical attributes of nodes/edges (e.g., weight, timestamp). |
| | Connectivity Logic | Connection rules between specific types of nodes (e.g., "nodes of type A must connect to nodes of type B"), negative connection rules (e.g., "nodes of type A must not connect to each other"). |
| **Domain-specific** | Synthetic | Graph model type (Complete, Star, ER Random, BA Scale-Free), full connectivity, degree of central node, edge probability p, number of attachment edges m. |
| | Molecule | Chemical valence rules (e.g., carbon is tetravalent), atom types and counts, chemical bond types and counts, presence of specific functional groups, ring structures (e.g., benzene ring) count and size, molecular weight, Quantitative Estimate of Drug-likeness (QED). |
| | Social Network | Community structure and count, degree of core nodes (high betweenness centrality), network diameter, average clustering coefficient, small-world effect, power-law distribution fit, assortativity/disassortativity. |
| | Transportation | Total number of nodes (intersections/sensors), edge directionality, count and connectivity of specific nodes (e.g., arterial roads), global reachability, network flow conservation, attributes like average speed/congestion index. |
| | IoT Security | Device types and counts, distribution of network protocols (e.g., HTTP, MQTT), normal/abnormal traffic patterns, communication frequency between specific devices, presence of unauthorized connections. |

GPU. All graph generation and processing were based on Python 3.12 and the NetworkX (v3.5) (**?**) library. For the instructional texts in the dataset, we employed a locally deployed Qwen3-8B model for language polishing.

**Prompt Engineering** To ensure a fair evaluation, we designed a unified zero-shot system prompt for all models, intended to purely test their generalization capabilities without any in-context examples. The prompt template is as follows:

```
You are a graph generation expert.  Your task is to
generate a graph based on the following description.
Output the graph in the format of a Python NetworkX
graph object.  Description: {instruction}
```

For tasks requiring direct JSON output, we adjusted the prompt accordingly to explicitly ask the model to generate a JSON string.

**Inference Parameters** To balance determinism and diversity in the generated outputs, all models utilized a uniform set of decoding parameters during inference: `temperature = 0.5`, `top_p = 0.9`, `top_k = 10`, and `max_tokens = 2048`.

**Evaluation Protocol** We constructed a fully automated evaluation pipeline that takes a model under evaluation and the Text2GraphBench dataset as input, performing the following steps:

1. Output Acquisition: Iterate through all test samples and invoke the model's API to obtain the output.

2. Output Parsing and Execution: We developed a robust parser that attempts to convert the model's textual output (e.g., adjacency lists, GML, Graphviz DOT) or its generated Python code into a standard NetworkX graph object. For code-based outputs, the parser executes the code in a secure sandbox environment and captures the returned graph object. Any output that results in a parsing failure, execution error, or timeout (set to 60 seconds) is marked as Invalid.

3. Constraint Verification and Metric Calculation: For each successfully parsed graph object, the evaluation module automatically loads the corresponding list of constraints and verifies them one by one. We define the core evaluation metric as the Constraint Pass Rate (CPR), calculated as:

$$\text{CPR} = \frac{\text{Total number of satisfied constraints}}{\text{Total number of constraints across all test samples}}.$$

# F INSTRUCTION TEMPLATE EXAMPLES

To ensure the diversity of instructions and to simulate real-world application scenarios, we designed a hierarchical system for generating instruction templates. All templates are formed by combining **general structural templates** with **domain-specific templates**, which are then populated with concrete constraint instances to generate the final natural language instructions.

## F.1 GENERAL STRUCTURAL TEMPLATES

These templates define the basic sentence structure of the instructions, covering various tones such as declarative, interrogative, and imperative.

- **Template 1 (Declarative):** "Please generate a graph that satisfies the following conditions: $\text{constraint}_1$, $\text{constraint}_2$, and $\text{constraint}_3$."
- **Template 2 (Requisitive):** "I need a graph to simulate..., which should have the characteristic of $\text{constraint}_1$ while also ensuring $\text{constraint}_2$."
- **Template 3 (Composite):** "Consider a graph where $\text{constraint}_1$. Additionally, the graph must not contain $\text{constraint}_2$. Finally, please set $\text{constraint}_3$ as a global property of the graph."

## F.2 DOMAIN-SPECIFIC TEMPLATES

These templates translate abstract constraint terminology into natural language descriptions specific to a particular domain.

- **Domain: Molecular Graph**
  - **Constraints:** `num_atoms(type='C', count=6)` & `has_ring(size=6)`
  - **Template:** "Generate a molecule containing 6 carbon atoms and ensure it forms a six-membered ring structure, just like a benzene ring".
- **Domain: Social Network**
  - **Constraints:** `num_communities=3` & `modularity > 0.4`
  - **Template:** "Construct a social network that can be clearly partitioned into 3 communities, and the modularity of the network must be greater than 0.4 to indicate a significant community structure".

By combining these two types of templates, we can programmatically generate tens of thousands of syntactically logical and content-rich instructions. This allows for a comprehensive and fine-grained evaluation of a model's language understanding and graph generation capabilities.

# G OTHER EXPERIMENTS AND DISCUSSIONS

**Generation Validity Analysis.** Before examining constraint pass rate, we first evaluate a more fundamental capability: whether models can generate valid and parsable graphs. Table 3 presents the validity rates of each model's outputs.

The experimental results demonstrate that top-tier models exhibit excellent performance in basic generation capabilities. Gemini-2.5-Flash achieves a parsing success rate of 99.76%, with a valid graph rate of 98.2%. DeepSeek-v3 and Qwen-Plus also perform exceptionally well, with parsing success rates exceeding 99.6%. However, smaller models exhibit clear performance stratification:

Llama4-Scout drops to 91.61%, GPT-OSS-20B achieves only 72.21%, and Qwen3-32B performs worst at 44.55%. This indicates that basic format control capability represents the primary obstacle faced by smaller models.

**Performance Analysis across Constraint Types.** We analyzed model performance across three constraint categories: structural constraints, semantic constraints, and domain-specific constraints. As illustrated in Figure 4:

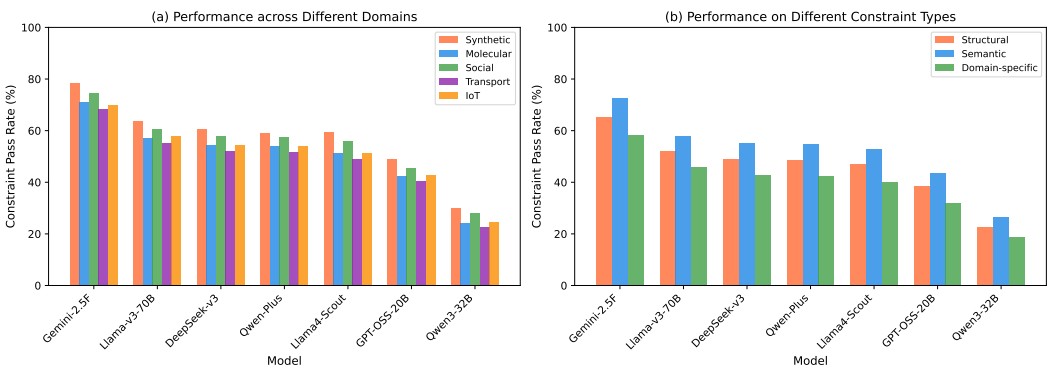

Figure 4: Model Performance Analysis across Different Constraint Types

Structural Constraint Analysis: Gemini-2.5-Flash demonstrates superior performance (65.2%), significantly outperforming Llama-v3-70B (52.3%) and other models. In node count control, Gemini-2.5-Flash achieves 69.25%, while most other models fall below 25%.

Semantic Constraint Analysis: Models generally perform better in this category, with Gemini-2.5-Flash reaching 72.8%, indicating that models are relatively proficient at handling node labels, attribute matching, and other semantic information.

Domain-Specific Constraint Analysis: Gemini-2.5-Flash achieves 58.3% on domain-specific constraints, while all other models remain below 45%, reflecting the limitations of models' domain knowledge.

**Performance Analysis under Varying Difficulty Levels.** We categorized problems into different difficulty levels based on the number of constraints and evaluated model performance accordingly. As shown in Figure 5:

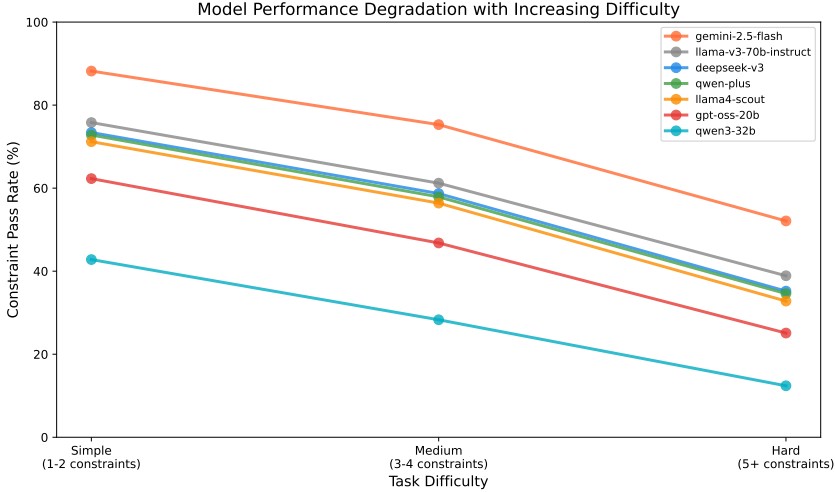

Figure 5: Model Performance Analysis under Different Difficulty Levels

Gemini-2.5-Flash exhibits the strongest robustness: 88.2% on simple tasks, 75.3% on medium difficulty, and 52.1% on challenging tasks. Llama-v3-70B degrades from 75.8% to 38.9%. Most notably, Qwen3-32B shows dramatic degradation from 42.8% to 12.4%, indicating that smaller models have short reasoning chains and struggle with multi-constraint tasks.

**In-depth Analysis of Specific Constraint Types.** Figure 6 presents detailed model performance on specific constraint types:

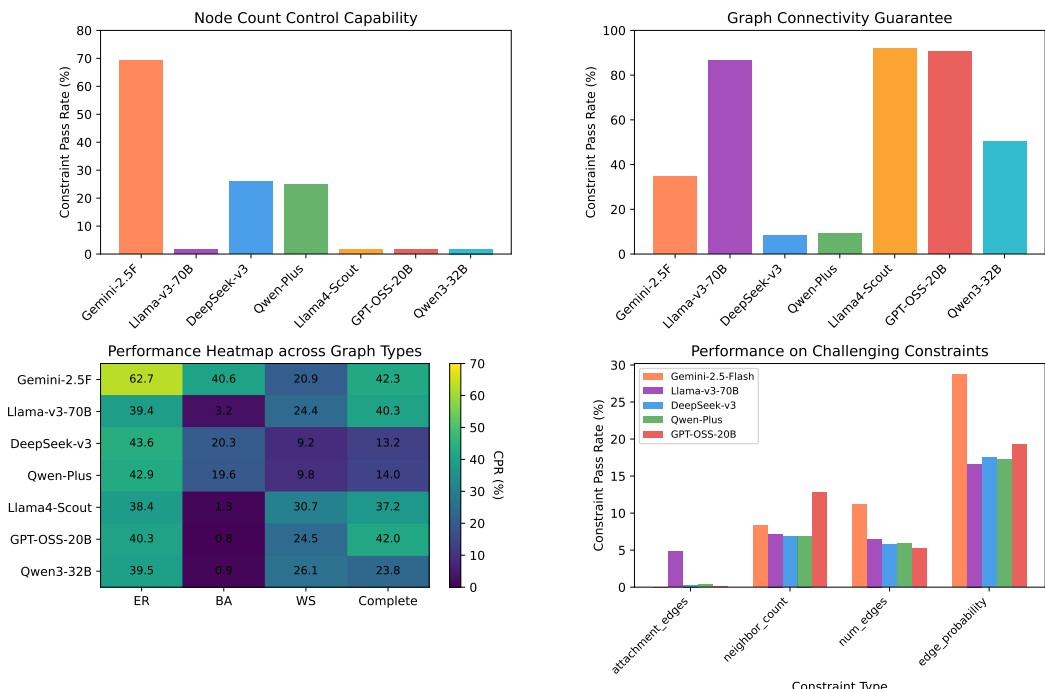

Figure 6: Detailed Analysis of Specific Constraint Types

Challenging Constraint Performance: Edge probability control is relatively easier (Gemini-2.5-Flash achieves 28.76%), while attachment edge control is most difficult (maximum 4.81%). Graph Type Specialization: Gemini-2.5-Flash performs best on Erdős-Rényi graphs (62.69%), but most models perform extremely poorly on Barabási-Albert graphs (¡5%), reflecting insufficient understanding of complex network generation mechanisms.

**Analysis of Intrinsic Limitations via Fine-tuning:** To address the critical question of whether the observed performance bottlenecks stem merely from a lack of training data or from intrinsic architectural limitations, we conducted Supervised Fine-Tuning (SFT) experiments. We selected two representative base models, Qwen3-8B and Llama3-8B. We constructed a diverse training set of approximately 20k pairs, covering both synthetic graphs (Erdos–Renyi, emphasizing global structural stochasticity) and real-world graphs (QM9, emphasizing strict chemical validity). Both models were fine-tuned using LoRA for 3 epochs.

As shown in Table 4, we observe a critical divergence in learnability across different constraint types. On one hand, semantic constraints prove to be highly learnable; for instance, Qwen3-8B achieved a significant gain of +7.4% in semantic alignment, suggesting that SFT effectively enables LLMs to map domain-specific terminologies to graph attributes. On the other hand, structural constraints remain a persistent bottleneck, with improvements being only marginal (+1.7% for Qwen3 and +3.1% for Llama3) despite the direct supervision. Consequently, the overall CPR improvement (ranging from 3% to 5%) is disproportionate to the substantial scale of high-quality training data provided.

These findings strongly suggest that the inability to enforce global topological consistency is an intrinsic limitation of current autoregressive LLMs. The "Structural Reasoning Wall" cannot be

easily bypassed simply by scaling up supervised data, further highlighting the unique value of Text2GraphBench as a diagnostic tool for architectural capabilities.

Table 4: Performance Improvement after Fine-tuning (CPR %).

| Model | Semantic Constraints | | | Structural Constraints | | | Overall CPR | | |
|---|---|---|---|---|---|---|---|---|---|
| | Pre-FT | Post-FT | $\Delta$ | Pre-FT | Post-FT | $\Delta$ | Pre-FT | Post-FT | $\Delta$ |
| Qwen3-8B | 21.3 | 28.7 | +7.4 | 12.4 | 14.1 | +1.7 | 18.5 | 22.1 | +3.6 |
| Llama3-8B | 15.1 | 21.4 | +6.3 | 7.8 | 10.9 | +3.1 | 13.8 | 19.4 | +5.6 |

## H  BROADER IMPACT AND USAGE SCENARIOS

**Significance to research communities:** Text2GraphBench serves as a critical bridge connecting the Graph and LLM communities. For graph research, it facilitates a paradigm shift from traditional statistical distribution fitting to semantic-driven generation, enabling precise control over graph topology via natural language. For the LLM community, it introduces a rigorous testbed for "structured constrained generation". By requiring models to map linear text to non-euclidean structures under hard constraints, our benchmark exposes intrinsic limitations in current LLMs' higher-order reasoning, serving as a valuable diagnostic tool for structural capabilities.

**Usage scenarios and future directions:** Beyond acting as a static evaluation set, Text2GraphBench functions as scalable infrastructure for future advancements. The high-quality instruction-graph pairs provide rich supervision signals for instruction tuning and pre-training to enhance LLMs' graph understanding. Furthermore, the verifiable CPR offers a deterministic and objective reward signal for Reinforcement Learning (RLHF). This enables researchers to train agents for complex scientific design tasks where structural validity is paramount, or to pinpoint specific architectural weaknesses in maintaining global topological consistency.

