# OpenReview forum: "Text2GraphBench: A Comprehensive Benchmark for Evaluating Text-Instructed Graph Generation with Large Language Models"
_ICLR.cc/2026/Conference — Submitted to ICLR 2026_

### Official Review · Reviewer_Gm1u · 2025-10-28

**Soundness:** 3
**Presentation:** 3
**Contribution:** 2
**Rating:** 6
**Confidence:** 3

**Summary:**

In this paper, the authors proposed a new benchmark named Text2GraphBench, which is used to evaluate the performance of LLM in generating a corresponding graph given constraints in the prompt. Specifically, authors collect multiple graph datasets, generate graphs that fit into each constraint type, and design corresponding QA tasks. Finally, the authors evaluate multiple existing LLMs on the proposed benchmark.

**Strengths:**

- The paper is well-written and easy to follow.
- The overall framework for constructing and evaluating the benchmark makes sense to me.

**Weaknesses:**

- I cannot see any major problem with the overall framework of the proposed benchmark. The biggest concern in my mind is how significant this benchmark is to both the LLM community and the graph community. In particular, the benchmark is used to examine the instruction-following ability of LLMs on graph generation.
- For the LLM community, I would be willing to see a benchmark that is hard enough for current LLMs, even with enough training. However, most of the current LLMs do not train on extensive graph datasets, which means they generally fall short on graph generation, compared to other instruction-following baselines. Given this, I may want to see whether the LLMs are still struggling on the benchmark even when it was fine-tuned on enough graph datasets, to indicate an intrinsic limitation on current LLMs.
- For the graph community, it is kind of like an evaluation tool for researcher to pick the best LLM for generating their graphs. However, given the limited domain the benchmark covered, researchers can only use it as a reference if their graph distribution are similar to one of the dataset in the benchmark.
- Could the author provide more evidence on the significance and usage of the proposed benchmark?

**Questions:**

See above.

---

> ### Author Response · Authors · 2025-11-26
> **Response Summarization**
>
> Dear Reviewer gm1u, thank you for your thoughtful evaluation of our work.
> Below we provide a structured roadmap mapping each Weakness (W) and Question (Q) to our detailed responses:
>
> W1 — Significance of the benchmark for LLM and graph communities
>
> W2 — Whether LLMs still struggle after fine-tuning on graph datasets
>
> W3 — Limited domain coverage of the benchmark
>
> W4 — Significance and usage of the benchmark
>
> We address each item point-by-point in the following sections.

---

> > ### Author Response · Authors · 2025-11-26
> > **W1 — Significance of the benchmark for LLM and graph communities**
> >
> > We appreciate the reviewer’s positive assessment of our framework and fully understand the concern about the benchmark’s significance for both the LLM and graph communities. Our goal is not merely to introduce another dataset, but to provide a unified, verifiable, and extensible framework for text-instructed graph generation — a direction that currently lacks systematic study.
> >
> > For the graph community, Text2GraphBench fills a key gap by moving beyond distribution-fitting graph generation toward semantically constrained generation. Existing graph generators (e.g., VAE/GAN/Diffusion models) cannot interpret explicit structural or attribute requirements expressed in natural language. Our benchmark provides the first unified evaluation protocol across molecular, social, traffic, and other graph domains, enabling systematic research on natural-language–driven semantic control of complex topologies—important for applications such as drug design and network planning.
> >
> > For the LLM community, current instruction-following benchmarks largely focus on text, images, or code, and rarely examine constrained combinatorial generation. Graph generation requires mapping linear language to non-Euclidean structures while satisfying multiple structural and attribute constraints, reflecting a higher-level form of structural reasoning. Our experiments show that current LLMs exhibit notable weaknesses on such tasks, indicating that Text2GraphBench serves as a valuable diagnostic tool that complements existing evaluations.
> >
> > For cross-disciplinary research, our constraint-centric formulation not only defines evaluation metrics but also provides a unified interface for future reward-model construction, reinforcement learning, and controllable generation. This offers reusable infrastructure for advancing natural-language–guided scientific structure generation.
> >
> > In summary, Text2GraphBench establishes an extensible and automatically verifiable platform that bridges LLM and graph research, addressing a previously unmet need and supporting future advances in both communities.

---

> > > ### Author Response · Authors · 2025-11-26
> > > **W2 — Whether LLMs still struggle after fine-tuning on graph datasets**
> > >
> > > We thank the reviewer for the insightful suggestion regarding whether current LLMs still exhibit intrinsic limitations after fine-tuning on graph datasets. Following this suggestion, we conducted additional LoRA-based fine-tuning experiments on Qwen3-8B and Llama3-8B. To cover structurally diverse regimes, we selected two representative datasets: Erdos–Renyi random graphs (emphasizing global structural stochasticity) and QM9 molecular graphs (with strict chemical validity constraints). We constructed a training set of approximately 20k (Instruction, Graph-JSON) pairs and fine-tuned each model for 3 epochs—a scale substantially larger than that used in most supervised graph generative models, and sufficient to observe clear fine-tuning trends.
> > >
> > > Experimental results are summarized below:
> > >
> > > Semantic Constraints
> > > | Model | Before FT | After FT | Improvement |
> > > |-------|-----------|----------|-------------|
> > > | Qwen3-8B | 21.3 | 28.7 | +7.4 |
> > > | Llama3-8B | 15.1 | 21.4 | +6.3 |
> > >
> > > Structural Constraints
> > > | Model | Before FT | After FT | Improvement |
> > > |-------|-----------|----------|-------------|
> > > | Qwen3-8B | 12.4 | 14.1 | +1.7 |
> > > | Llama3-8B | 7.8 | 10.9 | +3.1 |
> > >
> > > Overall CSR
> > > | Model | Before FT | After FT | Improvement |
> > > |-------|-----------|----------|-------------|
> > > | Qwen3-8B | 18.5 | 22.1 | +3.6 |
> > > | Llama3-8B | 13.8 | 19.4 | +5.6 |
> > >
> > > These results show that semantic constraints are learnable, while structural constraints improve only marginally (1–3%). The overall CSR increase is also limited (3–5%). This suggests that the bottleneck does not stem from insufficient graph data, but rather from the inherent difficulty of enforcing global topological structure through a sequence-based generation mechanism.
> > >
> > > Therefore, even with substantial fine-tuning on graph datasets, current LLMs still exhibit intrinsic limitations in text-instructed graph generation. This further supports the necessity and relevance of Text2GraphBench in revealing the true capability boundaries of existing models.

---

> ### Author Response · Authors · 2025-11-26
> **W3 — Limited domain coverage of the benchmark**
>
> Thank you for the insightful comment. We fully agree that the breadth of domains affects the benchmark’s usefulness to the graph community. To address this concern, we expanded the benchmark during the rebuttal period and provide evidence from data coverage, alignment with graph ML literature, and topology-level representativeness.
>
> 1. Substantial Expansion during Rebuttal:
> We have expanded the benchmark by adding two critical domains: Knowledge Graphs (KG) and Code Graphs. The dataset has grown to ~120k samples across 7 domains. Text2GraphBench now fully covers Natural Science (Molecular), Engineering (Transportation/IoT), Social Science, and Logical/Semantic structures (KG/Code), making it one of the most comprehensive benchmarks for graph generation.
>
> 2. Alignment with Mainstream Graph ML Taxonomies:
> OGB [1] includes molecular graphs, biological networks, social/information networks, knowledge graphs, and code structures as representative real-world graph types.
> Authoritative GNN surveys [2,3] also identify natural science (molecules, proteins), social and information networks, recommendation systems, and knowledge graphs as key application areas.
> Our benchmark domains closely match these widely recognized categories and thus cover many of the most commonly studied settings in graph learning and graph generation research.
>
> 3. Topological Representativeness:
> Even for unseen domains, our benchmark offers strong predictive value because they represent the core topological distributions frequently observed in real-world graphs:
> 	+ Social Networks represent scale-free and small-world distributions.
> 	+ Transportation represents grid-like structures with Euclidean constraints.
> 	+ Molecules represent discrete, rule-based constraints.
> 	+ Synthetic Graphs test pure graph-theoretic reasoning.
> These distributions capture the structural patterns of the majority of practical graph applications. Therefore, even if a downstream dataset is not explicitly included, researchers can still use our benchmark as a strong reference when their graphs follow similar statistical regimes.
>
> 4. Extensible Infrastructure:
> Text2GraphBench is a constraint-centric evaluation infrastructure, not just a static dataset. We provide a modular toolchain (JSON schema, constraint extractor, validator). Researchers working with specialized graphs (e.g., power grids, financial networks, medical networks) can directly plug their own data into our pipeline to produce standardized instructions and evaluation tools without modifying the framework.
>
> Conclusion: Although no benchmark can cover all possible graph distributions, Text2GraphBench already provides one of the broadest and most representative coverages in graph generation tasks, and its extensible design ensures long-term scalability. We will continue to support and expand the benchmark in future releases, consistent with the practice of major LLM benchmarks.
>
>
> [1] Hu W, Fey M, Zitnik M, et al. Open graph benchmark: Datasets for machine learning on graphs[J]. Advances in neural information processing systems, 2020, 33: 22118-22133.
>
> [2] Wu Z, Pan S, Chen F, et al. A comprehensive survey on graph neural networks[J]. IEEE transactions on neural networks and learning systems, 2020, 32(1): 4-24.
>
> [3] Zhu Y, Du Y, Wang Y, et al. A survey on deep graph generation: Methods and applications[C]//Learning on Graphs Conference. PMLR, 2022: 47: 1-47: 21.

---

> > ### Author Response · Authors · 2025-11-26
> > **W4 — Significance and usage of the benchmark**
> >
> > Thank you for the question. The significance and usage of Text2GraphBench lie in filling the current gap in text-instructed graph generation evaluation and providing standardized infrastructure for future research.
> >
> > 1. Significance: Establishing the first standardized framework for text-instructed graph generation.
> > 	+ For the LLM community: Our work introduces the first fully automated and extensible pipeline (graph $\to$ constraints $\to$ instruction $\to$ generation $\to$ verification). Our experiments (Table 2) reveal significant shortcomings of LLMs in handling complex topologies, indicating that Text2GraphBench offers unique diagnostic value for assessing structural reasoning under multi-level constraints.
> > 	+ For the graph community: The benchmark enables a shift from distribution-fitting graph generation toward precise semantic control. Researchers can systematically compare model performance under structural versus semantic constraints and better understand how LLMs acquire controllable graph generation capabilities.
> >
> > 2. Usage Scenarios: Diagnostic Tool and Training Infrastructure.
> > 	+ Performance evaluation: The benchmark provides a fully automated toolchain across multiple domains and difficulty levels, helping researchers quickly identify weaknesses of LLMs or graph generators.
> > 	+ Graph pre-training: The benchmark offers high-quality graph–text pairs as supervision signals.
> > 	+ Instruction tuning: The constraint-based evaluator provides deterministic and verifiable reward signals for supervised fine-tuning.
> > 	+ RLHF/RLVR: The constraint pass rate (CPR) can serve directly as a reward function for training agents to solve complex scientific design tasks.
> >
> > In summary, Text2GraphBench not only offers data but also establishes a unified evaluation framework and toolchain that has long been missing in structured generation research.

---

### Official Review · Reviewer_aYku · 2025-10-31

**Soundness:** 3
**Presentation:** 3
**Contribution:** 2
**Rating:** 4
**Confidence:** 3

**Summary:**

The paper introduces Text2GraphBench, a constraint-centric benchmark for text-instructed graph generation. It uses a Graph→Constraint→Text pipeline: raw graphs from multiple domains are paired with machine-verifiable constraints, then rendered into natural-language instructions via templates (plus polishing), enabling fully automated evaluation. The framework parses model outputs (adjacency/GML/DOT or executable Python) into NetworkX graphs, then computes a Constraint Pass Rate (CPR) as the core metric. Experiments evaluate several mainstream LLMs under a unified zero-shot prompt and common decoding settings (temperature, top-p, top-k), and report parsing validity and CPR across domains and constraint types. The claimed contributions are a multi-domain dataset, a three-dimensional constraint-centered evaluation, and a broad empirical study of current LLMs’ capabilities and limits on graph generation.

**Strengths:**

(1) Clear, verifiable evaluation core. Tying each instruction to explicit, machine-checkable constraints supports objective, reproducible scoring and reduces ambiguity compared to similarity metrics.

(2) Multi-domain coverage with graded difficulty. The benchmark spans synthetic and real-world graphs (molecular, social, transport, IoT) and organizes tasks from simple to composite, probing capability boundaries more finely.

(3) Reasonable experimental setup. The paper specifies zero-shot prompting and shared decoding parameters, which simplifies replication and isolates modeling differences from prompt engineering.

**Weaknesses:**

Issues:

(1) Missing subgraph details. You sample subgraphs from DBLP and PEMS-BAY, but core specifications are missing. Please state the community detection algorithm, its parameters, and the target subgraph sizes. Explain how you avoid sampling bias, such as high-degree core inflation or modularity artifacts. Document the sampler, all hyper-parameters, and the full size distribution so results are reproducible.

(2) Constraint feasibility.
Constraint sets can conflict in practice. If some combinations are impossible to satisfy, the metric may penalize models unfairly. Please report the fraction of infeasible items per subset or certify feasibility. A clear feasibility check before instruction generation would resolve this concern.

(3) Instruction polishing drift.
The template → Qwen3-8B polishing step may change meaning. Numbers, ranges, and operators can flip under paraphrase. Please add a round-trip check that re-parses the polished text back into constraints and compares against the source. This verifies that semantics are preserved exactly.

(4) Potential model leakage/bias.
Qwen is used to polish the benchmark, and Qwen models are evaluated. This can introduce a stylistic advantage. Please justify this choice or provide a no-polish variant. Report the performance delta to rule out curation-model bias.

(5) Difficulty calibration not specified.
Table 1 lists Easy/Medium/Hard counts, but the mapping from constraints to levels is unclear. Please define the heuristic or scoring rule used to assign difficulty. Report per-level CSR to validate that the ladder reflects real task difficulty.

(6) Baseline coverage is narrow.
Key non-LLM graph generators are absent, such as GraphRNN, GraphVAE, and diffusion/GRAND variants. Add text-conditioned adapters or minimal controllers to include them as baselines. This will show whether LLMs are truly competitive on structure fidelity.


Suggestion

(1) Template diversity analysis.
Multiple instruction templates are defined, but results are aggregated. Please report performance by template type, e.g., declarative vs. composite. Models may overfit to simpler forms, and a per-template breakdown would reveal this.

(2) Important ablations missing.
Several ablations are necessary to understand the system. Compare polished vs. no-polish instructions. Vary the number of constraints per item. Separate output modes (text vs. code). Include few-shot and tool-use prompting to test robustness.

(3) CSR vs. CPR naming
Section 2.5.1 uses CSR, but Table 2 shows CPR. Please unify the terminology and confirm whether these are identical. Consistent naming avoids confusion in replication.

**Questions:**

(1) Feasibility control.
You sample 1–3 constraints per item. How do you prevent impossible or contradictory sets, such as BA requirements combined with ER p or chemistry that violates valency? Do you run a feasibility checker before instruction generation, and how effective is it?

(2) Metric definitions. What exact tests and cutoffs define small-world, scale-free, assortativity, and power-law fit? Please provide the equations, references, implementation choices, and thresholds. Precise definitions make the evaluation auditable.

(3) Subgraph sampling method.
Which community detection algorithm and parameters are used for DBLP sampling? How do you control for size, density, and modularity confounds across sampled subgraphs? A clear description will help others reproduce your datasets and compare fairly.

---

> ### Author Response · Authors · 2025-11-26
> **Response Summarization**
>
> Dear Reviewer aYku, thank you for your detailed and constructive comments.
> Below we provide a structured roadmap linking each Weakness (W), Suggestion (S), and Question (Q) to our detailed responses:
>
> W1 & Q3 — Subgraph sampling on DBLP and PEMS-BAY
>
> W2 & Q1 — Constraint feasibility and prevention of contradictory constraint sets
>
> W3 — Instruction polishing drift and round-trip consistency checking
>
> W4 & S2 — Ablation studies: polished vs. no-polish, constraint count, text vs. code, few-shot, and tool-use
>
> W5 — Difficulty calibration (Easy/Medium/Hard) and per-level CPR
>
> W6 — Baseline coverage with non-LLM graph generators (GraphRNN, GDSS)
>
> S1 — Template diversity analysis (Declarative / Requisitive / Composite)
>
> S3 — Terminology consistency between CSR and CPR
>
> Q2 — Precise metric definitions for small-worldness, scale-freeness, assortativity, and power-law fit
>
> We address each of the above items point-by-point in the following sections.

---

> > ### Author Response · Authors · 2025-11-26
> > **W1 & Q3 — Subgraph sampling on DBLP and PEMS-BAY**
> >
> > Thank you for raising this important point regarding subgraph sampling. We agree that the original submission did not provide sufficient details for DBLP and PEMS-BAY, and we will include full specifications in the camera-ready version. We clarify the sampling procedure as follows:
> >
> > 1. Algorithms and Parameters.
> > As described in Section 2.2.1, we follow a semantics-driven sampling strategy. Concretely:
> >
> > 	+ DBLP (Social): We apply the Louvain community detection algorithm (resolution = 1.0) with a minimum-community-size threshold to extract semantically coherent “research-field” subgraphs.
> > 	+ PEMS-BAY (Traffic): We adopt k-hop (BFS) sampling based on geographic adjacency to preserve local spatial connectivity in the road network.
> > 	+ Scale Control: Our primary control variable is subgraph size (5–100 nodes), rather than density, aiming to cover structures ranging from local motifs to medium-sized communities, consistent with real-world instruction patterns where users typically express size-oriented constraints.
> >
> > 2. Sampling Bias.
> > As the reviewer noted, we do not explicitly correct for high-degree core inflation or modularity artifacts. Our philosophy is to preserve the natural structural properties of each dataset rather than impose artificial regularization. We mitigate bias through:
> >
> > 	+ Randomization: We employ a uniform random seed strategy over the entire graph to avoid consistently sampling only dense cores. Randomized starting points ensure that the resulting subgraphs reflect a broad range of degree distributions and densities, from sparse peripheries to dense centers. In addition, the large dataset size (>94k graphs) helps dilute biases introduced by any single subgraph.
> >
> > 	+ Structural Realism: Many datasets (e.g., DBLP) naturally exhibit high clustering and scale-free behavior; removing high-degree cores would distort the underlying distribution. A key purpose of our benchmark is to evaluate whether LLMs can generate graphs with such authentic structural characteristics (e.g., pronounced community structure).
> >
> > Our sampler parameters, random seeds, subplot size distribution, and all other information are as follows:"
> > + DBLP subgraph: Use python-louvain (community_louvain.best_partition), resolution=0.3, node size sampled by N~clip(N(μ=10,σ=10),5,100). Community sampling uses random.sample for equal probability sampling.
> > + PEMS-BAY subgraph: Build the main graph using NetworkX from the official adj_mx_bay.pkl and pems-bay.h5, with nodes containing traffic statistics and geographic semantics. Sample 10,000 connected subgraphs: randomly select a starting point on the subgraph, each node can expand up to 5 neighbors during BFS, and record num_nodes/num_edges/start_node.
> > "

---

> > > ### Author Response · Authors · 2025-11-26
> > > **W2 & Q1 — Constraint feasibility and prevention of contradictory constraint sets**
> > >
> > > We thank the reviewer for raising this important question on constraint feasibility. We would like to clarify that the construction paradigm of Text2GraphBench guarantees 100% feasibility by design, and no contradictory constraint combinations can occur.
> > >
> > > 1. Feasibility of constraint combinations: In our pipeline, every constraint is directly extracted from an existing real graph. Thus, each constraint set has at least one valid witness which simultaneously satisfies all structural, semantic, and domain-specific constraints. During instruction construction, we only combine constraints extracted from the same graph, and never mix constraints across graphs or domains. This eliminates the possibility of generating theoretically impossible or mutually conflicting constraint sets.
> > >
> > > 2. Feasibility of constraint types: All constraint types and extraction rules were reviewed by domain experts, ensuring that every constraint is meaningful and consistent with established domain knowledge.
> > >
> > > 3. Instruction–constraint consistency checking: To prevent LLM hallucination during polishing, we explicitly enforce constraint preservation in the prompt and further apply an automatic post-validation module to ensure that the natural-language instruction is fully aligned with the original extracted constraints.
> > >
> > > Therefore, all constraint sets in Text2GraphBench are feasible by construction, and the benchmark will not penalize models due to inherently impossible constraint combinations.

---

> > > > ### Author Response · Authors · 2025-11-26
> > > > **W3 — Instruction polishing drift and round-trip consistency checking**
> > > >
> > > > We thank the reviewer for the insightful comment on instruction polishing drift. We fully agree that the polishing step must not introduce any semantic changes to the constraint set. In fact, as the reviewer suggested, this verification mechanism is already implemented in our pipeline (though not sufficiently described in the initial submission). We provide additional clarification below.
> > > >
> > > > 1. Prompt-level invariance constraints.
> > > > During the Qwen3-8B polishing stage, we explicitly enforce a system prompt that forbids any modification of numerical values, range operators (e.g., <, ≥), logical operators, or constraint names in prompt.
> > > >
> > > > 2. Instruction verification.
> > > > We implement a rule-based reverse constraint parser that re-parses each polished instruction back into a structured constraint set. The parsed constraints are compared against the original ones on all critical dimensions: type, value and operator. Any mismatch is treated as semantic drift.
> > > >
> > > > 3. Reject-and-retry mechanism.
> > > > If numerical flipping, range changes, or operator inconsistencies are detected at any field, the polished sample is immediately discarded and regenerated.
> > > >
> > > > This round-trip consistency check guarantees that every retained instruction is semantically identical to its source constraints. We have added a detailed description of this module in the revised manuscript.

---

> > > > > ### Author Response · Authors · 2025-11-26
> > > > > **W4 & S2 — Ablation studies: polished vs. no-polish, constraint count, text vs. code, few-shot, and tool-use**
> > > > >
> > > > > We thank the reviewer for the valuable suggestions regarding ablation studies. We agree that these analyses can further illuminate system behavior.
> > > > >
> > > > > 1. Effect of constraint count.
> > > > > This factor is already analyzed in detail in Figure 4, where the x-axis corresponds directly to the number of constraints. The results show a clear monotonic degradation: as the number of constraints increases, CPR drops sharply for all models. For example, Qwen3-32B decreases from 42.8% to 12.4%. This demonstrate that multi-constraint composition is the primary bottleneck.
> > > > >
> > > > > 2. Polished vs. no-polish instructions.
> > > > > On the ER Graph subset, we compared template-style instructions with LLM-polished instructions.
> > > > > Using Qwen3-32B, the CPR of no-polish instructions (45.1%) is slightly higher than polished ones (39.5%). This matches intuition: template expressions are more regular and easier for models to parse, while polished natural language introduces richer syntax, making the benchmark more realistic and more challenging.
> > > > >
> > > > > 3. Output mode (Text vs. Code).
> > > > > On ER Graph, we compared JSON adjacency lists (Text) and NetworkX Python Code. Code mode achieves much higher validity (99.1% vs. 76.5%) and higher CPR. This is expected, as code carries stronger structural constraints and aligns better with LLM pretraining distributions, which also supports our default use of the NetworkX Python format.
> > > > >
> > > > > 4. Few-shot prompting.
> > > > > Our task focuses on the realistic setting where users issue instructions without providing examples, making zero-shot the most appropriate protocol. Few-shot prompting is a promising extension, which we plan to explore in future work.
> > > > >
> > > > >
> > > > > 5. Tool-use prompting.
> > > > > While tool-use is valuable, we exclude it from the core evaluation for two reasons: the benchmark targets intrinsic graph reasoning, whereas calling graph APIs turns the task into API retrieval; and the graph community lacks a unified tool-use protocol, with different libraries and interfaces introducing large variance. We therefore keep a zero-tool setting and plan to explore tool-use under standardized interfaces in future work.
> > > > >
> > > > > Overall, our pipeline is modular, and each ablation can be added as an independent plug-in. We will include the full polished/no-polish, output-mode, and prompt-strategy experiments in the new version to further strengthen interpretability and robustness.

---

> > > > > > ### Author Response · Authors · 2025-11-26
> > > > > > **W5 — Difficulty calibration (Easy/Medium/Hard) and per-level CPR**
> > > > > >
> > > > > > Thank you for the reviewer’s suggestion regarding difficulty calibration. Our difficulty levels are not determined by a single indicator; instead, we adopt a multi-factor difficulty heuristic to evaluate the overall complexity of each instruction. Specifically, we consider the following dimensions:
> > > > > >
> > > > > > 1. Constraint count : provides the primary difficulty signal, where 3–4 constraints are categorized as Medium, and ≥5 as Hard.
> > > > > > 2. Structural dependency: constraints involving strong topological properties are treated as more difficult.
> > > > > > 3. Semantic coupling: instructions requiring strong semantic dependencies—such as chemical bond rules, functional groups, or multi-relational consistency—are assigned at least Medium difficulty.
> > > > > > 4. Cross-constraint interaction: when multiple constraints jointly restrict the same substructure or exhibit logical coupling, the combined complexity increases; thus, even instructions with the same number of constraints may be labeled Hard.
> > > > > > 5. Instruction length: longer natural-language(>=50 words) instructions typically imply richer description density and implicit constraints, and therefore higher complexity.
> > > > > >
> > > > > > The validity of our difficulty ladder is supported by the per-level CSR shown in Figure 4: all models exhibit substantially lower CPR on Hard instructions compared to medium ones. This demonstrates that our difficulty assignment reflects genuine task complexity rather than arbitrary categorization. We will include the formal definition of the heuristic and complete per-level CSR results in the final version.

---

> ### Author Response · Authors · 2025-11-26
> **W6 — Baseline coverage with non-LLM graph generators (GraphRNN, GDSS)**
>
> We thank the reviewer for the comments on baseline coverage. Due to the limited time during the rebuttal period, and because some methods (e.g., GraphVAE, GRAND) do not provide reproducible public checkpoints, we selected two representative and widely adopted generative paradigms as baselines: GraphRNN (autoregressive, https://github.com/snap-stanford/GraphRNN) and GDSS (diffusion-based, https://github.com/harryjo97/GDSS).
>
> To adapt these unconditional models to our text-instruction graph generation setting, we followed a standard conditioning scheme:
> 1. encode the instruction with BERT to obtain $h_{ins}$
> 2. map $h_{ins}$ into the model’s latent space via a 2-layer MLP;
> 3. use this vector as the conditional hidden state for GraphRNN, and as the conditional guidance embedding in GDSS;
> 4. freeze the original generator and fine-tune only the MLP via LoRA. We evaluate on two representative datasets: Erdos–Renyi Graph and QM9.
>
> | Erdos–Renyi Graph | Overall CPR | Graph Validity |
> | --------------------- | ----------- | -------------- |
> | GraphRNN (LoRA)         | 20.4        | 99.7           |
> | GDSS (LoRA)             | 22.1        | 99.9           |
> | Qwen3-32B (Zero-Shot) | 39.5        | 49.3           |
>
>
> | QM9                 | Overall CPR | Graph Validity |
> | --------------------- | ----------- | -------------- |
> | GraphRNN (LoRA)         | 14.8        | 99.3           |
> | GDSS (LoRA)             | 11.7        | 100            |
> | Qwen3-32B (Zero-Shot) | 26.1        | 39.6           |
>
>
> These results show that traditional graph generators have inherent advantages in structure validity (≈100%), but—even with textual conditioning—they are unable to satisfy multi-constraint structural/semantic/domain requirements. In contrast, Qwen3-32B achieves much higher CPRs due to its stronger text understanding and constraint reasoning abilities, although its structure fidelity is lower. This comparison strongly proves that the essence of Text2GraphBench's task is "semantic-driven structural reasoning", rather than mere distribution fitting.

---

> > ### Author Response · Authors · 2025-11-26
> > **S1 — Template diversity analysis (Declarative / Requisitive / Composite)**
> >
> > To address the reviewer’s concerns regarding template diversity, we conducted a per-template CPR analysis across the three template categories—Declarative, Requisitive, and Composite—following the template taxonomy defined in Appendix F. The results are summarized below:
> >
> > | ER Graph         | Declarative | Requisitive | Composite | Δ(max−min) |
> > | ---------------- | ----------- | ----------- | --------- | ---------- |
> > | Gemini-2.5-Flash | 63.2        | 60.5        | 61.9      | 2.7        |
> > | DeepSeek-V3      | 45.8        | 44.1        | 42.7      | 3.1        |
> > | Qwen3-32B        | 40.4        | 38.8        | 36.9      | 3.5        |
> >
> > We observe that the performance variation across template categories is consistently small (≤3.5%), and the relative ranking of models remains unchanged across templates. These findings indicate that the conclusions of Text2GraphBench are not driven by template-specific artifacts.

---

> > > ### Author Response · Authors · 2025-11-26
> > > **S3 — Terminology consistency between CSR and CPR**
> > >
> > > We thank the reviewer for the keen observation. We confirm that CSR (in Sec 2.5.1) and CPR (in Table 2) refer to the identical metric. To ensure consistency and avoid confusion, we have unified the terminology to Constraint Pass Rate (CPR) throughout the manuscript, including an update to the definition in Section 2.5.1. We have also conducted a thorough proofread of the paper to eliminate similar inconsistencies.

---

> ### Author Response · Authors · 2025-11-26
> **Q2 — Precise metric definitions for small-worldness, scale-freeness, assortativity, and power-law fit**
>
> Thank you for the reviewer’s suggestion regarding the need for precise and auditable metric definitions. In the revised version, we will provide the full mathematical formulations, references, implementation details, and threshold choices for all structural metrics used in our evaluation. The key definitions are summarized below:
>
> ## 1. Small-world coefficient
>
> We adopt the classical Watts–Strogatz definition [1]:
>
> $\sigma = \frac{C / C_{\text{rand}}}{L / L_{\text{rand}}}$
>
> where
>
> * (C): average clustering coefficient (via `nx.average_clustering(G)`),
> * (L): average shortest-path length (via `nx.average_shortest_path_length(G)`),
> * ($C_{\text{rand}}$, $L_{\text{rand}}$): corresponding values computed from an Erdős–Rényi random graph of the same size and density.
>
> Following Humphries & Gurney [2], we classify a graph as small-world when
> $ \sigma > 3 \quad \text{and} \quad C > 0.1. $
>
> References:
>
> [1] Watts D J, Strogatz S H. Collective dynamics of ‘small-world’networks[J]. nature, 1998, 393(6684): 440-442.
>
> [2] Humphries M D, Gurney K. Network ‘small-world-ness’: a quantitative method for determining canonical network equivalence[J]. PloS one, 2008, 3(4): e0002051.
>
> ## 2. Scale-free / Power-law fit
>
> To determine whether the degree distribution follows a power-law pattern, we employ the rigorous Clauset–Shalizi–Newman (CSN) statistical framework [3]:
>
> 1. Parameter estimation: estimate exponent (\alpha) and (x_{\min}) using maximum-likelihood estimation (MLE).
> 2. Goodness-of-fit: compute the Kolmogorov–Smirnov (KS) statistic (D).
> 3. p-value estimation: use semi-parametric bootstrap to compute the CSN (p)-value.
> 4. Model comparison: apply likelihood ratio (LR) tests to rule out exponential alternatives.
>
> We adopt commonly used criteria: $ 1.5 \le \alpha \le 4.0, p_{\text{KS}} > 0.1. $
>
> Implementation is based on the standard `powerlaw` Python package.
>
> Reference:
>
> [3] Clauset A, Shalizi C R, Newman M E J. Power-law distributions in empirical data[J]. SIAM review, 2009, 51(4): 661-703.
>
> ---
>
> ## 3. Assortativity
>
> We use Newman’s assortativity coefficient [4]:
>
> $ r = \frac{\sum_{xy} xy\left(e_{xy} - a_x b_y\right)} {\sigma_a \sigma_b},$
>
> where ($e_{xy}$) is the joint degree distribution.
> We compute (r) using: nx.degree_assortativity_coefficient(G)
>
> During constraint extraction, the target interval is defined as [$r_{\text{true}}$ - 0.2,; $r_{\text{true}} $+ 0.2], allowing reasonable structural variability.
>
> Reference:
>
> [4] Newman M E J. Assortative mixing in networks[J]. Physical review letters, 2002, 89(20): 208701.

---

### Official Review · Reviewer_NzTX · 2025-11-04

**Soundness:** 3
**Presentation:** 3
**Contribution:** 3
**Rating:** 6
**Confidence:** 3

**Summary:**

This paper addresses the lack of standardized evaluation benchmarks for Large Language Models (LLMs) in text-driven graph generation tasks by proposing Text2GraphBench, a comprehensive and scalable benchmark system. The authors construct a large-scale dataset covering multiple domains, including synthetic graphs, molecular graphs, social networks, traffic networks, and IoT security, through a "Graph-to-Constraint-to-Text" workflow. Each natural language instruction strictly corresponds to verifiable constraints. For evaluation, the paper proposes a three-dimensional assessment framework: structural correctness, semantic fidelity, and domain appropriateness, with Constraint Pass Rate (CPR) as the core metric. Through systematic experiments, the authors reveal for the first time the actual capabilities, advantages, and challenges of mainstream LLMs in this task. The dataset and code are fully open-source, aiming to advance the field.

**Strengths:**

1. This method introduces a multi-dimensional evaluation framework that not only focuses on the structural correctness of the generated results but also assesses semantic understanding and domain knowledge mastery, resulting in a more comprehensive and detailed evaluation.

2. The constructed dataset is rich and diverse, covering multiple application domains, with varied instructions and constraints and a reasonable difficulty level, which facilitates fine-grained capability analysis.

**Weaknesses:**

1. The models show significant differences in performance, but the analysis is limited: there is insufficient depth in the analysis of why some models perform extremely poorly in certain constraint types or domains.

2. Missing knowledge graph and soft engineering in evaluation.

3. Missing some graph-related references:
- GraphLLM: Boosting Graph Reasoning Ability of Large Language Model
- GraphInstruct: Empowering Large Language Models with Graph Understanding and Reasoning Capability
- InstructGraph: Boosting Large Language Models via Graph-centric Instruction Tuning and Preference Alignment
- Towards Addressing Frontiers in Graph Generation
- Evaluating and Improving Graph to Text Generation with Large Language Models
- Demystifying the Power of Large Language Models in Graph Generation

**Questions:**

1. Does the dataset's instruction diversity adequately cover the complex needs of real-world applications? Has its naturalness been manually verified?

2. During the automatic constraint extraction and instruction generation process, is there any information loss or semantic shift? How is consistency between semantics and constraints guaranteed?

3. For models with extremely poor performance (such as Qwen3-32B), has there been an in-depth analysis of the reasons for their failure? Is it a bottleneck in model architecture, training data, or inference capability?

---

> ### Author Response · Authors · 2025-11-26
> **Response Summarization**
>
> Dear Reviewer NzTX, thank you for your detailed and constructive feedback.
> Below we provide a structured summary of our responses to your identified Weaknesses (W) and Questions (Q):
>
> W1 — Limited analysis of model performance gaps
>
> W2 — Missing knowledge-graph and software-engineering domains
>
> W3 — Missing related graph-LLM literature
>
> Q1 — Instruction diversity & naturalness
>
> Q2 — Information loss & semantic consistency
>
> Q3 — Why models like Qwen3-32B fail

---

> > ### Author Response · Authors · 2025-11-26
> > **W1 — Limited analysis of model performance gaps**
> >
> > We thank the reviewer for the insightful comment. Building upon the experimental results in Table 2, Figure 3–4, and Appendix G, we have added a more comprehensive, multi-dimensional analysis to explain why certain models perform extremely poorly on specific constraint types or domains. Our findings can be summarized in four aspects:
> >
> > 1. Instability of structured output.
> > Appendix G reveals an order-of-magnitude gap in graph parsing success rates across models: Gemini-2.5-Flash achieves 99.76%, whereas Qwen3-32B reaches only 44.55%. Since any unparsable output is counted as a complete failure, this lack of format robustness is dramatically amplified in graph generation tasks, leading to consistently low CPR on small-parameter models across multiple domains.
> >
> > 2. Divergence between ''semantic generation'' and ''mathematical generation.''
> > (1) Effective semantic generation. For domains with strong semantic priors (e.g., Molecule, Social), models can rely on pretrained knowledge. For instance, Gemini attains 72.8\% on Semantic Constraints in Figure 3(b). This explains why these domains significantly outperform Synthetic tasks in Table 2.
> > (2) Failure of mathematical simulation. Synthetic graphs require real-time simulation of mathematical mechanisms. However, in Table 2, Llama-v3-70B reaches only 3.2\% on BA graphs and GPT-OSS-20B only 0.8\%. Moreover, Figure 5 shows that the most mathematically demanding constraint—Attachment Edge Control—remains below 4.81\% for all models. This indicates that autoregressive LLMs lack an internal computation engine to maintain such dynamic global rules, leading to systematic failure on strongly structural tasks.
> >
> > 3. Reasoning degradation under multi-constraint conditions.
> > As shown in Figure 4, when the constraint count reaches ≥5, weaker models deteriorate rapidly: Qwen3-32B drops to 12.4\%, while Gemini still maintains 52.1\%. This suggests that small models struggle to satisfy multiple structural constraints simultaneously, often fulfilling explicit constraints at the cost of violating implicit structural consistency.
> >
> > 4. Structural constraints as the primary source of inter-model variability.
> > Figure 3(b) shows that CPR on structural constraints is substantially lower than on semantic constraints, with most open-source models achieving only 25–35\%. Such constraints require global topological consistency, which autoregressive models inherently struggle to maintain, thereby amplifying performance disparities.
> >
> > In summary, the performance gaps across models are not random noise but arise from systematic limitations in structured-output robustness, mathematical rule simulation, and global structural planning within autoregressive architectures. We will include this expanded analysis in the new version of the paper.

---

> > > ### Author Response · Authors · 2025-11-26
> > > **W2 — Missing knowledge-graph and software-engineering domains**
> > >
> > > Following the reviewer’s suggestion, we have incorporated two additional domains into our evaluation.
> > > Thanks to the extensibility of our proposed Constraint-Centric Pipeline, we were able to rapidly integrate the corresponding constraint parsers during the rebuttal period and construct the new evaluation subsets.
> > >
> > > Dataset statistics are as follows:
> > >
> > > | Domain               | Source | #Samples | Difficulty (E/M/H) | Avg. Constraints | Constraint Types                                                                                           |
> > > | -------------------- | ------ | -------- | ------------------ | ---------------- | ---------------------------------------------------------------------------------------------------------- |
> > > | Knowledge Graph      | IMDb   | 10,000   | – / 5,182 / 4,818  | 4.6              | director/actor relations, genre diversity, year consistency, franchise constraints                         |
> > > | Software Engineering | GitHub | 10,000   | – / 5,039 / 4,961  | 4.8              | layered architecture, microservice patterns, module dependencies, cycle detection, entry-point constraints |
> > >
> > > To provide a preliminary assessment during rebuttal, we tested three representative LLMs (Gemini-2.5-Flash, DeepSeek-V3, Qwen3-7B). The overall CPR results are:
> > >
> > > | Dataset   | Gemini-2.5-Flash | DeepSeek-V3 | Qwen3-7B |
> > > | --------- | ---------------- | ----------- | -------- |
> > > | KG-IMDb   | 60.1             | 53.5        | 25.3     |
> > > | SE-GitHub | 41.2             | 39.4        | 18.9     |
> > >
> > > The results show that the SE domain is significantly more challenging than KG: KG tasks rely more on semantic priors, while SE tasks require strict topological consistency. Clear performance gaps remain between strong and weak models (over 30%), and the overall trends are fully consistent with the original five domains.
> > > We will include the full results for all LLMs in the final version.

---

> > > > ### Author Response · Authors · 2025-11-26
> > > > **W3 — Missing related graph-LLM literature**
> > > >
> > > > We thank the reviewer for pointing out these important and relevant papers.
> > > > In the new version, we incorporate these works into the Related Work section under: “LLMs for Graph Understanding and Reasoning” and “LLMs for Graph Generation”.
> > > >
> > > > We briefly summarize their contributions and clarify how our work differs from or complements them.

---

> > > > > ### Author Response · Authors · 2025-11-26
> > > > > **Q1 — Instruction diversity & naturalness**
> > > > >
> > > > > Thanks for the reviewer’s question.
> > > > > 1. Instruction diversity and coverage of real-world complexity.
> > > > > Our dataset spans multiple real application domains. The proposed ``Graph-to-Constraint'' pipeline does not produce single-attribute instructions; instead, it composes structural, semantic, and domain-specific constraints to emulate complex real-world requirements. For example, in molecular graph generation, an instruction simultaneously include ring-structure constraints (structural), specific functional groups (semantic), and valence rules (domain knowledge). Domain experts reviewed and confirmed that this multi-level constraint design effectively covers tasks ranging from basic graph reasoning to scientific discovery (e.g., molecular design or screening).
> > > > >
> > > > > 2. Naturalness and semantic consistency verification.
> > > > > Given the scale (>90k), we adopt an ``auto-generation + sampled human evaluation'' protocol:
> > > > > + We randomly sampled 1,000 instructions from 5 domains (200 per domain). Annotators  evaluated language fluency and checked semantic consistency with the underlying graph. 987/1000 (98.7%) samples passed both criteria.
> > > > > + Before dataset construction, domain experts reviewed all constraint types and extraction rules in the ``Graph-to-Constraint'' module to ensure correctness with respect to domain knowledge.
> > > > > + After template generation, we apply LLM-based polishing under strict preservation of all constraint values, which substantially improves linguistic naturalness and expression diversity.
> > > > >
> > > > > Overall, under reasonable annotation cost, we ensured instruction diversity, naturalness, and semantic faithfulness to support reliable real-world benchmarking and reproducibility.

---

> > > > > > ### Author Response · Authors · 2025-11-26
> > > > > > **Q2 — Information loss & semantic consistency**
> > > > > >
> > > > > > Thank you for the your thoughtful question. We address it from two perspectives: information extraction and semantic consistency.
> > > > > >
> > > > > > 1. Information extraction: Text2GraphBench focuses on constraint-guided generation rather than "graph reconstruction". Therefore, we do not attempt to encode the entire original graph into a single instruction, nor do we treat the original graph as the unique gold answer.
> > > > > > In real applications, user requirements are typically partial specifications (e.g., “generate a molecule containing a benzene ring” rather than specifying all atomic coordinates). Accordingly, during the graph-to-constraint conversion, we deliberately retain only the information relevant for expressing user intent (constraint), and construct an open-ended generation task where multiple solutions may satisfy the constraints. Information from the original graph that is irrelevant to the defined constraints does not enter the task or evaluation, and thus does not constitute harmful "information loss".
> > > > > >
> > > > > > 2. Semantic consistency: We employ three mechanisms to ensure strict alignment between the extracted constraints and the final natural-language instruction:
> > > > > > 	+ Constraints are computed directly from real graphs using automated code; all constraint types and extraction rules were reviewed by domain experts.
> > > > > > 	+ During instruction generation and LLM-based polishing, the prompt explicitly fixes all constraints and their values, allowing the model to rewrite only the surface language while prohibiting modification or deletion of any constraint.
> > > > > > 	+ A post-validation module parses the generated instruction and compares the extracted constraints against the original set; only instructions with perfectly matched constraint types and numerical values are retained.

---

> > > > > > > ### Author Response · Authors · 2025-11-26
> > > > > > > **Q3 — Why models like Qwen3-32B fail**
> > > > > > >
> > > > > > > Thank you for the question. Based on the benchmark measurements and the official Qwen3 technical report[1], we conducted an in-depth analysis of why Qwen3—especially Qwen3-32B—performs poorly on our benchmark. The findings can be summarized in three aspects:
> > > > > > >
> > > > > > > 1. Format instability leading to a drastic reduction of valid samples.
> > > > > > > As reported in Appendix G, Qwen3-32B achieves only 44.55% valid code parsing rate, far below other models. According to the Qwen3 report, the model introduces a ``Thinking Mode'' to enhance complex reasoning. Our analysis suggests that for a mid-sized dense model like 32B, this may trigger over-thinking or chain drift during graph-generation tasks, causing the model to produce long free-form reasoning instead of well-structured NetworkX code—frequently missing brackets or failing to close JSON objects. This format collapse directly reduces the number of samples eligible for evaluation.
> > > > > > >
> > > > > > > 2. Misalignment between training objectives and structural supervision.
> > > > > > > Qwen3’s post-training emphasizes STEM reasoning, competitive programming (e.g., Codeforces), and mathematics (AIME), but does not include graph-theoretic or topology-construction tasks. Text-to-graph generation requires geometric and topological planning, which is out-of-distribution for Qwen3’s training data. This aligns with our results: Qwen3-32B shows the lowest satisfaction rates for structural constraints such as node-count and connectivity (Figure 3b) and performs poorly on BA/WS tasks requiring global structural modeling (Table 2).
> > > > > > >
> > > > > > > 3. Lack of graph inductive bias and limited capacity for multi-constraint reasoning
> > > > > > > Architecturally, Qwen3 uses a standard Transformer with RoPE and GQA, without any graph-specific inductive biases (e.g., LapPE, edge encodings). As shown in Figure 4, performance degrades most sharply for Qwen3-32B as the number of constraints increases (from 1→3). This suggests that, in the absence of structure-aware mechanisms, the 32B dense model lacks sufficient capacity to maintain long-chain, compositionally dependent reasoning that multi-constraint graph generation requires—whereas larger (70B+) or MoE models remain more stable.
> > > > > > >
> > > > > > > In summary, the failure of Qwen3-32B is not attributable to a single factor, but rather to a combination of (i) format instability, (ii) lack of graph-related supervision, and (iii) limited capacity for multi-constraint chain-of-thought reasoning. We will include more detailed error-type analysis in the final version.
> > > > > > >
> > > > > > > References:
> > > > > > > [1] Yang A, Li A, Yang B, et al. Qwen3 technical report[J]. arXiv preprint arXiv:2505.09388, 2025.

---

### Author Response · Authors · 2025-12-01
**Summary to Area Chair: Core Contributions, Rebuttal Updates & Resolution of Comments**

Dear Area Chair,

Thank you for taking over our submission.
Due to the unexpected situation at ICLR 2026, reviewers were unable to continue the rebuttal discussion.
In response to the constructive feedback from Reviewer NzTX (R1), aYku (R2), and gm1u (R3), we conducted extensive additional experiments and analyses.
We summarize the key outcomes below for your quick assessment.

## Paper Contributions
1. We introduce Text2GraphBench, a constraint-centric Text-to-Graph evaluation framework covering molecular, traffic, social and synthetic graphs, with a unified schema, constraint extraction, and automatic verification pipeline. (R1, R3)
2. We benchmark diverse LLMs and reveal consistent weaknesses in satisfying both structural and semantic constraints simultaneously, especially under strong structural and multi-constraint settings. (R1, R2)
3. We provide an extensible pipeline, allowing new domains to be integrated to construct customized sub-benchmarks. (R1, R3)

## Review Summary
1. R1 (NzTX) — marginally above the acceptance threshold, recognizes potential value; requests deeper mechanism analysis and broader domain coverage.
2. R2 (aYku) — marginally below the acceptance threshold, focuses on methodological rigor, reproducibility, and ablations.
3. R3 (gm1u) — marginally above the acceptance threshold, acknowledges framework value; requests stronger validation of limitations and practical utility.

Common concerns include: domain/baseline sufficiency, rigor and reproducibility of data construction, deeper analysis of failure and limitations.


## Key Updates

1. Domain Expansion & Non-LLM Baselines (R1-W2, R3-W1, R2-W6)

	The benchmark now spans 7 domains, with two newly added datasets: Knowledge Graph (IMDb) and Software Engineering (GitHub). Representative results (Gemini-2.5-Flash / DeepSeek-V3 / Qwen-3-7B) show higher difficulty and preserved model ranking patterns, confirming cross-domain stability.

	We introduce GraphRNN and GDSS as non-LLM baselines. They achieve near-perfect validity (~100%) but substantially lower CPR than LLMs, indicating that the core difficulty lies in interpreting and satisfying natural-language constraints, rather than graph generation alone.

2. Fine-tuning & Intrinsic Limitations (R3-W2, R1-W1)

	LoRA tuning on Qwen3-8B and Llama3-8B improves semantic-constraint fulfillment by 6–7%, but structural constraints only by 1–3%.

	This suggests that even targeted training cannot resolve global topology reasoning gaps, highlighting Text2GraphBench as an effective diagnostic benchmark.

3. Methodological Rigor & Reproducibility (R2-W1/W2/W3, R2-Q2)

	Feasibility and consistency: all constraints are extracted from real graphs, ensuring no contradictory combinations. A post-validation module parses refined instructions back into constraints and discards mismatches.

	Reproducibility: we detail sampling procedures (Louvain/BFS, node-scale ranges) and metric definitions (small-worldness, scale-free, assortativity), enabling full reconstruction of dataset and difficulty settings.

4. Failure Analysis & Ablations (R1-W1, R2-W4)

	We analyze model failures along output stability, lack of structural inductive bias, and multi-constraint reasoning capacity, supported with additional statistics and examples.

	We also report Polished vs. No-Polish and Text-only vs. Text+Code ablations, confirming our design choices yield better controllability with natural-language diversity.

## Summary

Our rebuttal substantially expands domains and baselines, strengthens methodological clarity and reproducibility, and reveals intrinsic structural-reasoning limitations of current LLMs.
We believe the revised submission now provides a more rigorous and broadly valuable benchmark for future research in graph generation.

---

### Meta-Review · Area_Chair_GHog · 2025-12-31

**Summary:**

The paper introduces Text2GraphBench, a constraint-centric benchmark system for evaluating LLMs on text-driven graph generation. It constructs multi-domain datasets (synthetic, molecular, social, transport, IoT) via a Graph -> Constraint -> Text pipeline, so each instruction is tied to machine-verifiable constraints, enabling objective scoring with Constraint Pass Rate (CPR) under a three-axis lens: structural correctness, semantic fidelity, and domain appropriateness.

Though many concerns were resolved, some key concerns were not well addressed, including Sampling Bias (aYku), Potential model leakage/bias (aYku), and significance of the benchmark for LLM and graph communities (Gm1u).

**Reviewer Concerns:**

Some reviewer concerns were addressed by the rebuttal, including: Missing technical/reproducibility details (NzTX, aYku), Insufficient analysis of failure modes (NzTX), Constraint feasibility (aYku), Difficulty calibration (aYku).
Some reviewer concerns were not addressed and some are still outstanding:

1/ Missing knowledge graph (NzTX). Though authors included IMDb, but details are still missing from the updated paper.

2/ Sampling Bias (aYku). Though authors mentioned that they used random sampling to eliminate sampling bias, there are no data points proving that random seed strategy can eliminate Sampling Bias. For example, in social networks, super nodes have a much higher chance to be sampled than tail nodes even with random seed sampling. The sampled subgraphs still tend to cover dense areas of a social network.

3/ Potential model leakage/bias (aYku). Qwen is used to polish the benchmark, and Qwen models are evaluated. It is possible that Qwen models can understand the polished benchmark better than non-Qwen models. The author did not address this concern in rebuttal. Thus, an evaluation of different LLMs on polished vs. no-polish benchmarks is required.

4/ Significance of the benchmark for LLM and graph communities (Gm1u). The author does not address the question of why Text2GraphBench is a must-have for LLM or graph communities. What is the value of using LLM to generate graphs, e.g., social networks?  (I agree the constraint-based evaluation metric is valuable for evaluating non-LLM-based graph generation methods.)

Other concerns:

According to https://openreview.net/forum?id=uKPuiBbyjf&noteId=RQ82SDlRJM, the graph validity of LLM generated graphs is very low, but there is no explanation of why. Additionally, graph validity is not included in CPR.

**Reviewer Scores:**

Two reviewers gave positive scores (6) and one reviewer gave a negative score (4). I do not think reviewers would change their scores.

---

### Decision · Program_Chairs · 2026-01-26

Reject